# On the effectiveness of Rotation-Equivariance in U-Net: A Benchmark for Image Segmentation

**Robin Ghyselinck**                                     *robin.ghyselinck@unamur.be*
*Faculty of Computer Science*
*NaDI institute*
*University of Namur*
*Rue Grandgagnage 21, 5000 Namur, Belgium*

**Valentin Delchevalerie**                               *valentin.delchevalerie@unamur.be*
*Faculty of Computer Science*
*NaDI & naXys institutes*
*University of Namur*
*Rue de Bruxelles 61, 5000 Namur, Belgium*

**Bruno Dumas**                                          *bruno.dumas@unamur.be*
*Faculty of Computer Science*
*NaDI institute*
*University of Namur*
*Rue Grandgagnage 21, 5000 Namur, Belgium*

**Benoît Frénay**                                        *benoit.frenay@unamur.be*
*Faculty of Computer Science*
*NaDI institute*
*University of Namur*
*Rue Grandgagnage 21, 5000 Namur, Belgium*

**Reviewed on OpenReview:** *https://openreview.net/forum?id=UcrVnXBdZI*

## Abstract

Numerous studies have recently focused on incorporating different variations of equivariance in Convolutional Neural Networks (CNNs). In particular, rotation-equivariance has gathered significant attention due to its relevance in many applications related to medical imaging, microscopic imaging, satellite imaging, industrial tasks, etc. While prior research has primarily focused on enhancing classification tasks with rotation equivariant CNNs, their impact on more complex architectures, such as U-Net for image segmentation, remains scarcely explored. Indeed, previous work interested in integrating rotation-equivariance into U-Net architecture has focused on solving specific applications with a limited scope. In contrast, this paper aims to provide a more exhaustive evaluation of rotation equivariant U-Net for image segmentation across a broader range of tasks. We benchmark their effectiveness against standard U-Net architectures, assessing improvements in terms of performance and sustainability (i.e., computational cost). Our evaluation focuses on datasets whose orientation of objects of interest is arbitrary in the image (e.g., Kvasir-SEG), but also on more standard segmentation datasets (such as COCO-Stuff) so as to explore the wider applicability of rotation equivariance beyond tasks undoubtedly affected by rotation equivariance. The main contribution of this work is to provide insights into the trade-offs and advantages of integrating rotation equivariance for segmentation tasks.

## 1 Introduction

Recently, many studies have focused on integrating different forms of equivariance in Convolutional Neural Networks (CNNs) (Weiler & Cesa, 2019; Delchevalerie et al., 2023; Worrall et al., 2017; Cohen & Welling, 2016; Kumar et al., 2018). This idea is largely inspired by the superior ability of CNNs to solve image-related tasks (e.g., classification, detection, segmentation) compared to fully-connected architectures. Indeed, CNNs inherently leverage translation equivariance (i.e., shifts in position) by extracting features through the use of convolutions that implement weight sharing across translations. This property is particularly desirable for solving image processing tasks. This sharing of weights leads to models that are more efficient in image processing, both in terms of accuracy and computational resources as compared to their fully-connected counterparts.

In previous work on introducing equivariance in CNNs, rotation equivariance has attracted particular attention. As a matter of fact, rotation equivariance is highly relevant for many applications, especially when the overall orientation of the image is arbitrary. Such cases arise in many fields, including medical imaging (Oreiller et al., 2022; Elaldi et al., 2024), microscopic imaging (Chidester et al., 2019b; Graham et al., 2020), satellite imaging (Marcos et al., 2018; Li et al., 2020) and industry 4.0 (Marcos et al., 2016). In these cases, the overall orientation of the images often contains no meaningful information. Instead, this orientation adds noise, potentially hindering the training process of a model. As a result, constraining CNNs with rotation equivariance can incorporate prior knowledge into the model, thereby improving its performance thanks to a reduction in unnecessary variability.

When applied to relevant tasks, rotation equivariant CNNs can outperform their vanilla counterparts (Weiler & Cesa, 2019; Worrall et al., 2017; Delchevalerie et al., 2023). Specifically, equivariant CNNs often achieve higher accuracy or at least state-of-the-art performance while using fewer parameters. This leads to an improvement from a sustainability point of view, since training models with fewer parameters reduces the overall computational cost. However, only a few studies have evaluated the benefits of these techniques in more complex CNN architectures or on tasks beyond classification (Chidester et al., 2019a; Mitton & Murray-Smith, 2021; Oreiller et al., 2022). Furthermore, while it is well-established that rotation equivariance is beneficial for specific applications that involve arbitrary orientations, it may also prove to be useful for more general tasks that do not inherently require global rotation invariance *a priori*. In fact, learning rotation equivariant features can still help in solving such tasks more effectively.

In this work, we explore the impact of rotation equivariance on segmentation tasks, in particular using the popular U-Net (Ronneberger et al., 2015) architecture. While most prior research has focused on evaluating the benefits of equivariant CNNs for classification tasks, we aim to fill the gap by benchmarking their performance on the more complex task of image segmentation. We assess whether introducing rotation equivariance can improve model accuracy, reduce the computational resources required by the model, or even both. Moreover, we also evaluate equivariant models for tasks where the orientation of the image is not necessarily arbitrary or not explicitly related to the task, such as objects segmentation (e.g., building or ship) in aerial imagery, where such objects maintain consistent orientations relative to the ground. For example, recognizing buildings or ships in aerial images. This is mainly motivated by the fact that learning rotation invariant features can still be useful for such applications, where the performance of an equivariant model is higher. This justifies the use of standard segmentation datasets that focus on general tasks (such as the well-known COCO-Stuff by Lin et al. (2014)) along with other datasets that are more linked to the concept of rotation invariance and focus on specific, narrow tasks (e.g., Kvasir-SEG by Jha et al. (2020), a dataset for polyp detection in colonoscopy images). In this work, we provide a comprehensive analysis of the trade-offs and advantages of incorporating rotation equivariance into U-Net, thereby contributing to a broader understanding of its applicability across a range of use cases. The PyTorch implementation of our work is publicly available as an open-source code on Github[1].

Our key findings are that the effectiveness of equivariant U-Nets compared to vanilla U-Nets is highly contingent on both object characteristics and segmentation task complexity. In datasets where objects can have arbitrary shapes and do not have a semantically meaningful orientation, equivariant models often outperform vanilla architectures. Conversely, for rotationally symmetric objects, vanilla architectures prove superior.

---

[1]https://github.com/RobGhys/seg-equi-architectures

One explanation is that the rotation equivariance constrains the filters and reduces their expressiveness, hindering the learning process of the model. For general-purpose segmentation, the performance metrics are similar between equivariant and vanilla U-Nets. However, for those general tasks, the performance gap arises in small data regimes, where equivariant models are consistently worse than the vanilla ones. This observation highlights how low data availability affects the trade-off between architectural constraints and model expressiveness.

Section 2 provides background knowledge on equivariance in CNNs, as well as on the U-Net architecture. Subsequently, Section 3 presents the related work on using equivariant models for segmentation tasks. Then, the datasets and models considered in assessing the usefulness of equivariant U-Nets are presented in Section 4, while the experiments and results are presented in Section 5. Finally, Section 6 presents a discussion on the results before concluding in Section 7.

## 2 Background

This section provides background knowledge about the task of image segmentation and the U-Net architecture. Furthermore, it presents the current state-of-the-art techniques that allow for the implementation of equivariance in CNNs.

### 2.1 Image Segmentation

Image segmentation is a branch of computer vision that aims at separating an image into meaningful objects or regions. Typically, four tasks can be considered. First, semantic segmentation (see for example Long et al. (2015)) where each pixel in the image is assigned to a specific category based on what stuff it represents (e.g., a level of blue for a bird, a level of yellow for a giraffe, white for the clouds, etc.). Although it is a subset of semantic segmentation, we specifically mention the task of binary segmentation (Kittler & Illingworth, 1986), when each pixel is either background (the black pixel) or foreground, i.e., the object of interest (the white pixel). Second, instance segmentation (see for example, He et al. (2017)), extending semantic segmentation by also making the distinction between individual instances within an object class, answering the questions of how many objects there are in an image, and what are their extents (e.g., each of the three giraffes is uniquely identified in the image). Third, panoptic segmentation (Kirillov et al., 2019), where all the objects in the scene, as well as the remaining stuff are labeled. This goes beyond semantic segmentation and instance segmentation. Fourth, dense pose estimation, where pixels that belong to people are labeled, together with their body parts (body, head, locations of limb and attitude, see for example Felzenszwalb & Huttenlocher (2005) and Andriluka et al. (2014)). This work only deals with semantic segmentation tasks.

Many segmentation methods have been developed in the last decades, prior to the introduction of deep learning. To name a few, thresholding-based approaches, such as a method by Otsu (1979) that segments images based on pixel intensity values. Edge detection methods, such as the Canny edge detector (Canny, 1986), identify boundaries between different image regions. Region-based segmentation techniques, including region growing (Adams & Bischof, 1994) and watershed transformation (Beucher & Mathmatique, 2000), segment images by grouping pixels with similar properties. Clustering-based techniques, such as Mean Shift (Comaniciu & Meer, 2002), deal with segmentation by iteratively shifting data points toward high-density regions in feature space. Next, graph-based methods, such as Normalized cuts (Shi & Malik, 2000) and graph cuts (Boykov & Jolly, 2001), represent the image as a graph where pixels are nodes and edges connect neighboring pixels. Finally, energy minimization or Bayesian inference techniques, like the Texton-Boost system (Shotton et al., 2007) uses shared boosting to use information from the color distributions of images, information about the location and texture of the object.

With the advent of deep learning, those methods fell out of favor because they relied on the use of hand-crafted features, often requiring a specific domain knowledge or expertise, as well as extensive fine-tuning for each particular task (Garcia-Garcia et al., 2017). Conversely, deep learning methods have the ability to learn feature representations for the pixel labeling problem in an end-to-end fashion.

## 2.2 Image Segmentation with Deep Learning

Deep learning has revolutionized image segmentation, offering models that learn hierarchical features directly from data. Popular architectures include Fully Convolutional Networks (FCNs) (Long et al., 2015), Encoder-Decoder architectures, such as U-Net (Ronneberger et al., 2015) or DeepLab (Chen et al., 2018), pyramids networks, and transformer-based models.

FCNs are the first architecture that made it possible to semantically label each pixel with one neural network. This method replaces fully connected layers with convolutional layers such that the spatial information is preserved (Long et al., 2015). Although this work showed interesting results, the resolution permitted by the technique is low, i.e., the boundaries of the objects are not well defined. Other works added conditional random fields (Zheng et al., 2015) at the end of the network to deal with the problem of poor resolution. Building on this work, DeepLab (Chen et al., 2018), improves the model's performance with the introduction of atrous convolutions in addition to the conditional random fields.

Another technique, U-Net, introduced for biomedical segmentation tasks, features an encoder-decoder architecture with fine-level connections that help recover spatial information lost during a downsampling phase (Ronneberger et al., 2015).

Some architectural variations rely on the feature pyramid network (Lin et al., 2017) and use top-down connections to help semantic information flow to higher-resolution maps (see for example the Pyramid Scene Parsing Network by Zhao et al. (2017)).

In recent years, transformer-based architectures, such as SETR (Zheng et al., 2021), Swin-Transformer (Liu et al., 2021) and SegFormer (Xie et al., 2021), have gained attention because of their ability to model long-range dependencies thanks to the attention mechanism. However, transformer-based models generally require large datasets for training because of their lack of inductive biases (Dosovitskiy et al., 2021). Conversely, CNN-based methods, such as U-Net, with its strong inductive biases and efficient use of labeled data, are better suited for those data settings. Furthermore, the incorporation of rotation-equivariant modifications to CNNs provides additional benefits in scenarios where rotational invariance is desirable.

## 2.3 U-Net

Assigning a label to each pixel in an image is essential in various domains such as medical imaging, autonomous driving, and satellite imagery, where understanding the spatial structure of objects within an image is necessary for predicting segmentation masks. Among these models, U-Net (Ronneberger et al., 2015) has emerged as one of the most popular and effective architectures for image segmentation. This model was originally developed for biomedical image segmentation and has since been widely adopted across different fields. The model has a 'U' shape and is based on an encoder-decoder structure with skip connections (see Figure 1), where the encoder is responsible for extracting hierarchical features and the decoder focuses on reconstructing segmentation masks. The ability of U-Net to combine low-level spatial details with high-level semantic information makes it particularly effective for boundary delineation at the pixel-level, which allows for producing segmentation masks for objects of arbitrary shapes.

The encoder aims at progressively extracting useful information at different scales in the input image. It is responsible for capturing the context of the input image through a series of downsampling convolutional layers. Each layer in the encoder path applies a set of convolutional operations followed by activation functions and pooling operations. This process progressively reduces the spatial dimensions of the input, while increasing the number of feature channels, effectively condensing the image information into a much more compact representation. At the bottom of the U-shape (after the encoder), one finds the bottleneck layer. It acts as the bridge between the encoder and decoder, and consists of convolutional operations that further distill the compact representation of the image. This allows the network to capture complex and abstract patterns within the input data. The decoder then aims to reconstruct an output image with the initial spatial dimensions. It employs transposed convolutions (also known as up-convolutions) to upsample the feature maps, gradually restoring the spatial dimensions that were reduced during the encoding process. A core feature of the U-Net architecture is its skip connections between the encoder and decoder layers. When the input data moves downward the encoder (e.g., a $224 \times 224$ pixels image), it undergoes successive

downsampling operations that increase the receptive field and capture abstract features. Unfortunately, the corollary is that fine spatial details are lost. Nonetheless, the skip connections address this limitation by connecting each encoder layer to its corresponding decoder layer (i.e., there is a one-to-one mapping between each encoder and each decoder layer) in order to concatenate their feature maps. This allows high-resolution spatial information to flow from the early encoder layers (at the top of the U-Net) to the decoder. Moreover, the abstract features learned at the bottom of the U-Net (i.e., in the bottleneck) are preserved. In medical image segmentation for instance, these connections help preserve boundary details and anatomical structures that might otherwise be lost during the encoding process.

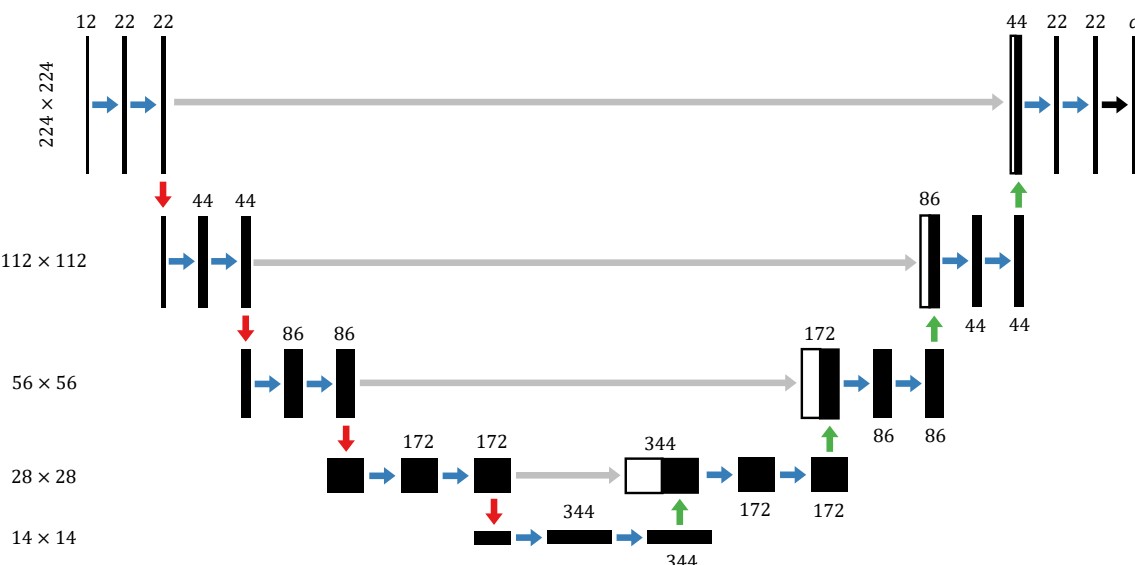

Figure 1: The U-Net architecture for the large vanilla U-Net (adapted from Ronneberger et al. (2015)). It illustrates the encoder, bottleneck, and decoder paths with skip connections. The input image is preprocessed by two convolutional layers (not represented in this illustration). Red arrows correspond to downsampling, blue arrows to convolutions and green arrows to upsampling. The final images obtained after the last black arrow are the $c$ segmentation masks for the $c$ corresponding classes.

The U-Net architecture is particularly effective in scenarios with limited training data, which is common in medical imaging and other specialized fields. The extensive use of data augmentation, along with the skip connections that help preserve spatial information, allows U-Net to produce precise segmentation maps even when trained on small datasets. Furthermore, its ability to work end-to-end makes it a preferred choice for segmentation tasks that require detailed, pixel-level accuracy.

## 2.4 Equivariance in CNNs

Recently, numerous studies have focused on incorporating equivariance in CNNs (see for example Elaldi et al. (2024); Delchevalerie et al. (2023); Mitton & Murray-Smith (2021); Oreiller et al. (2022) or Graham et al. (2020)). Rotation equivariance, in particular, has attracted significant attention due to its relevance in image recognition. The property of rotation equivariance is very important in several domains, like medical, satellite, and microscopy imaging where the orientation of objects can be arbitrary, meaning that any variation in the orientation of the object should not affect its classification or detection (Weiler & Cesa, 2019). This means that some images do not have a preferred global orientation. For example, this applies to a building in satellite imagery, where regardless of the viewing angle from above (i.e., aerial view), it is still the same object. Medical scans are other concrete examples, where a tumor should be recognizable regardless of the patient's position in the machine during the capture of the images.

There are three categories of methods that deal with rotation equivariance: (i) those that enhance rotation robustness without providing formal guarantes of equivariance, (ii) those that offer some mathematical guarantes, but only for a discrete set of rotations, such as those defined by the cyclic $C_n$ and dihedral $D_n$ groups, and (iii) those that give mathematical guarantees of equivariance for continuous transformations, primarily involving the special orthogonal $SO(2)$ or the orthogonal $O(2)$ groups.

The most widely adopted technique from the first category is data augmentation (Quiroga et al., 2018). The main idea of data augmentation is to artificially increase the size of the training set by incorporating variability in the data that is unrelated to the considered task. In the context of rotation equivariance, one may introduce rotated versions of the images in the training dataset, for example by rotating the entire image by any multiple of 90°. However, while robustness can be considerably increased thanks to the use of data augmentation, it is obvious that the model will try to learn the rotation equivariance from training examples instead of being constrained to be rotation equivariant. Unfortunately, this does not lead to any formal guarantee of equivariance and has severe limitations from an efficiency point of view.

Regarding the second category, the most famous method is probably group-CNN by Cohen & Welling (2016). In group-CNN, filters are artificially duplicated in several versions with the application of all the transformations from the considered discrete symmetry group. For example, if the group $C_4$ (all the rotations for $\alpha \in \{0°, 90°, 180°, 270°\}$) is considered, the final number of filters will be multiplied by four after performing 90°, 180° and 270° rotations on the initial filters. In other words, filters are learned in the same fashion as for vanilla CNN. Yet, when forwarding an input image, each filter is artificially duplicated according to the transformations contained in the considered symmetry group and convolved with the image. At the end of the method, a symmetry pooling is performed to summarize the activations of the different variants of the same filters. Only one copy of each filter needs to be updated during the back-propagation. They are thus defined as real trainable parameters.

Finally, for the third category, general-$E(2)$-equivariant CNNs ($E(2)$-CNNs) by Weiler & Cesa (2019) have become the primary method, achieving equivariance to continuous groups through a finite set of irreducible representations. The idea is closely related to that of steerable filters (Cohen & Welling, 2017; Weiler et al., 2018; Graham et al., 2020), which is widespread in the equivariant CNNs literature. Regarding general-$E(2)$-equivariant CNNs, they build on the fact that any representation of a finite or compact group can be decomposed into a direct sum of irreducible representations. This decomposition in irreducible representations can be used afterwards to constrain the filters. This idea has the advantage of being very general. However, the authors report that irreducible representation for $SO(2)$ or $O(2)$ generally does not lead to better performance than the discrete versions $C_n$ or $D_n$.

In this work, we focus on the group-CNN (Cohen & Welling, 2016) method from the second category. This choice is motivated by the fact that methods from the first one do not provide any guarantees on the property of equivariance. With these methods, one cannot be confident that any trained model fully relies on rotation equivariant features. Methods from the third category, like general-$E(2)$-equivariant CNNs, are not selected either. This exclusion is based on previous results in the literature showing that approximating the continuous equivariance with a discrete one is generally a good solution, and the use of groups (such as $C_4$) is less computationally expensive (Weiler & Cesa, 2019; Delchevalerie et al., 2023). Consequently, we choose to use the publicly available and general implementation of $E(2)$-CNNs[2] that facilitates building group-CNNs by leveraging discrete symmetry groups.

## 3 Related Work

Exploiting rotation equivariance has been extensively studied for many classification tasks. A non-exhaustive subset of those application domains includes medical imaging (Oreiller et al., 2022; Elaldi et al., 2024), microscopic imaging (Chidester et al., 2019b; Graham et al., 2020), satellite imaging (Marcos et al., 2018; Li et al., 2020) and industry 4.0 (Marcos et al., 2016). However, a limited number of works try to address segmentation tasks while applying rotation equivariance constraints.

---

[2]https://github.com/QUVA-Lab/e2cnn

To the best of our knowledge, the first study to consider exploiting rotation equivariance in U-Net architecture is Marcos et al. (2016), which introduced Rotation Equivariant Vector Field Networks (RotEqNet), an approach for incorporating rotation equivariance into CNNs through vector field representations. Their method encodes orientation information explicitly within the network architecture by implementing specialized filters that respond to features at multiple orientations and aggregating these responses into vector fields. The authors show that their architecture results in compact models in terms of parameters size, and is able to achieve comparable performance to those of much larger networks. However, this work is limited to the ISBI 2012 2D EM segmentation challenge (Arganda-Carreras et al., 2015), a specialized microscopy binary segmentation dataset. This method was fine-tuned for this specific application, and it is not clear how RotEqNet would perform on different or more complex (e.g., higher resolution, importance of global orientation in the image, etc.) data or tasks (e.g., semantic segmentation task).

The work by Weiler et al. (2018) proposed a general framework for constructing rotation equivariant CNNs using steerable filters. They developed a method where convolutional kernels transform predictably under rotations. Thanks to their approach, the network processes rotated inputs with consistency. By encoding this geometric prior knowledge into the network architecture, they demonstrated substantial parameter efficiency while maintaining or improving performance compared to conventional CNNs. In addition, the authors provide theoretical guarantees of equivariance and practical implementation strategies for their steerable filter basis through circular harmonic decomposition. Similarly to the work by Marcos et al. (2017), this study is limited to the ISBI 2012 2D EM segmentation challenge dataset (Arganda-Carreras et al., 2015).

A semantic segmentation model developed by Winkens et al. (2018) and Linmans et al. (2018) uses rotation and reflection symmetries to improve sample efficiency and robustness to image transformations. This work is an extension of the group equivariant CNN that is able to deal with segmentation tasks thanks to an equivariant $(G \rightarrow \mathbb{Z}_2)$-convolution that transforms feature maps from a group to planar representations. The authors evaluate their work on a tumor localization dataset.

Another work by Chidester et al. (2019a) investigates the use of group convolutions (restricted to the use of the C4 group) in a U-Net architecture for nucleus segmentation in histopathological images. The authors conclude that using equivariance to rotations of 90° "enables the network to learn better parameters that generalize well to simple transformations, namely, translations and rotations".

Other works have considered the use of rotation equivariance, such as the work by Mitton & Murray-Smith (2021), group-CNNs are used for the very specific task of deforestation segmentation. Oreiller et al. (2022) introduce a specific locally rotation invariant bispectral U-Net for the task of multi-organ nucleus segmentation.

Having considered those previous works, we believe that integrating rotation equivariance into CNNs remains under-explored and has to be further researched, more particularly for segmentation tasks using U-Net architectures. The few previous works only investigate one particular configuration for rotation equivariance, and only tackle one specific task. The current literature lacks insights and thorough evaluations of the performance trade-offs and sustainability assessment in terms of computational costs involved in incorporating rotation equivariance into U-Net for segmentation across diverse datasets. The aim of this work is to close this gap by delivering a more comprehensive assessment of the effectiveness of rotation equivariant U-Net models across various segmentation tasks, while shedding more light on the use of rotation equivariant constraints. By evaluating both rotation-sensitive and standard datasets, we not only investigate the usefulness of rotation equivariance for tasks inherently linked to orientation arbitrariness, but also consider its potential advantages for segmentation tasks that are more general.

## 4 Datasets and Models

This work aims to assess the effectiveness of equivariant U-Net models under several practical conditions: nature of the selected task, dataset size, number of classes for the segmentation task, and model size (number of trainable parameters). First, one can train U-Net models on datasets dealing with tasks of different nature. Either those tasks involve a notion of rotation invariance at global or local scales (e.g., polyp segmentation mask) or they do not (e.g., segmentation mask of a pedestrian). Second, the selection of datasets is performed

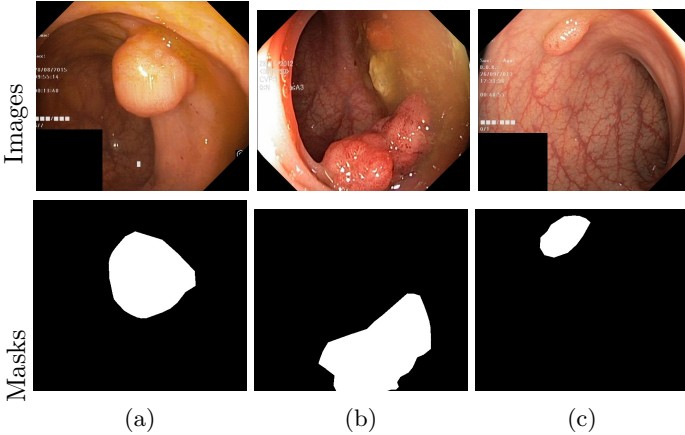

Figure 2: Example of colonoscopy images with their segmentation mask from the Kvasir-SEG dataset. One can notice that the polyps vary in shape and size, and are located at different positions within the images.

such that smaller and larger data regimes exist. Indeed, the usefulness of equivariant constraints may depend on the quantity of available data for training (Delchevalerie et al., 2023). For this reason, in addition to the use of datasets of different sizes, we also consider using different amounts of data for each dataset. Third, both binary (one foreground class and one background class) and semantic segmentation (involving a small to large number of distinct classes) are considered. Last, one can consider smaller and larger U-Net models in terms of number of trainable parameters. This is motivated by the idea that equivariant models are constrained by prior knowledge, which may reduce their required number of parameters to obtain a model with similar performance (Cohen & Welling, 2016). In agreement with the aforementioned prerequisites, this section presents the datasets as well as the different architectures considered in this work.

## 4.1 Datasets

Five datasets are considered in this work. The first three datasets are designed for binary segmentation tasks, i.e., masks only contain a foreground and a background. These datasets are Kvasir-SEG (Jha et al., 2020), NucleiSeg (Janowczyk & Madabhushi, 2016), and URDE (De Silva et al., 2023). Two additional datasets are selected for semantic segmentation, namely COCO-Stuff (Caesar et al., 2018) and iSAID (Waqas Zamir et al., 2019). While the COCO-Stuff is much larger in size than the datasets from the first category, iSAID has the same order of magnitude in terms of image counts. In addition, COCO-Stuff and iSAID are much more general than the first three and are often used as baseline datasets for benchmarking segmentation models. Further, the number of classes that are featured in these two last datasets is greater than in the first three datasets. Also, some datasets feature objects for which the orientation does not matter (like a polyp in Kvasir-SEG), as opposed to other datasets (e.g., a train in COCO-Stuff). Consequently, these five datasets cover a large spectrum of segmentation tasks: smaller/larger datasets, binary/semantic segmentation, and rotation invariant/standard applications. Table 1 presents a summary of key information about each dataset, i.e., the number of images that it contains, images resolution, and the intended use case for the dataset. The next part of this section provides more practical information about them.

**Kvasir-SEG**   This first dataset (Jha et al., 2020) contains endoscopic images. Its focus is on the segmentation of polyps in human colons. It contains $1,000$ images with varying sizes, from $625 \times 513$ pixels for the smallest image to $1920 \times 1072$ pixels for the largest one. Each image is assigned to a binary mask that shows the items of interest, i.e., the polyps. Figure 2 presents three examples of paired images and masks in the dataset. This dataset is expected to be very informative on the ability of a model to learn from only a few images. Furthermore, the task is inherently rotation invariant, as polyps may appear with any arbitrary orientation.

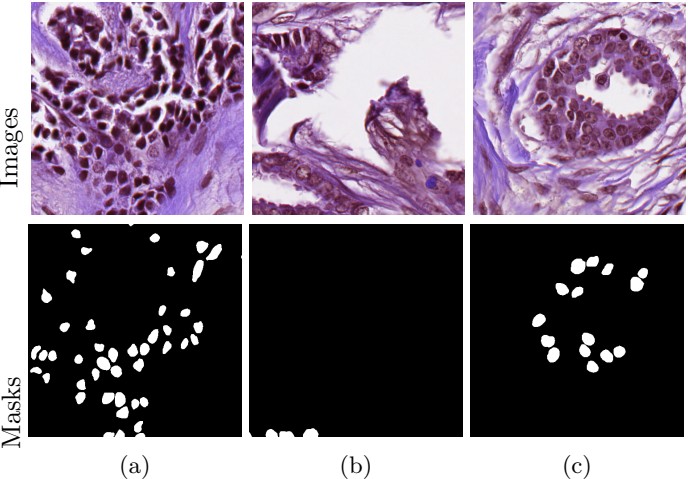

Figure 3: Example of images with their segmentation mask from the NucleiSeg dataset. Notice the multiplicity of nuclei in single patched images.

**NucleiSeg**   This second dataset (Janowczyk & Madabhushi, 2016) is designed for segmenting cell nuclei in histopathological images. It contains 135 images with multiple nuclei per image. Each image has a resolution of $2,000 \times 2,000$ pixels, much higher than the resolution of images from the other binary segmentation datasets considered in this work. Because of this, each image is pre-processed to generate a set of lower resolution images by extracting random patches of $448 \times 448$ pixels within the original images and masks. Because there are multiple nuclei in each image, the pre-processing will generate sub-images containing multiple nuclei too, such that the images remain useful for the segmentation task. For each image-patch pair, this operation is repeated 30 times to ensure diversity and comprehensiveness. Figure 3 illustrates a few paired image-mask examples resulting from the pre-processing. The NucleiSeg dataset is considered to be informative on the performance of segmentation models in biomedical images having many foreground items per image. The orientation of the object is not expected to contain any information and objects of interest always have a circular shape.

**URDE**   The name of this third dataset, URDE (De Silva et al., 2023), stands for Urban Roadside Environment and is the last involving binary segmentation. The dataset focuses on detecting dust clouds induced by vehicles traveling on unsealed roads. It includes $7,000$ images of $1,024 \times 1,024$ pixels with their annotation masks. For illustration, Figure 4 presents a sample of paired images and masks in the dataset. In this case, the problem is more general and orientation may be seen as a particular form of information.

**COCO-Stuff**   This fourth dataset (Caesar et al., 2018) is a large-scale dataset commonly used for general segmentation tasks. It includes over $164,000$ images with 91 stuff categories (plus one class unlabeled) as annotation masks, providing a diverse and extensive set of images for evaluating general-purpose segmentation algorithms. The resolutions vary between images, with a maximum of $640 \times 640$ pixels. Figure 5 presents a few examples of paired images and masks in the dataset. The complexity and size of the dataset make it a must-have for benchmarking semantic segmentation tasks. As with the URDE dataset, orientation in COCO-Stuff contains meaningful information rather than being arbitrary.

**iSAID**   This fifth and last dataset (Waqas Zamir et al., 2019) is based on the DOTA (Xia et al., 2018) dataset. While the latter focuses on object detection in satellite imagery, the former focuses on segmentation of the same images. iSAID consists of $2,806$ images with annotations for 15 different object classes such as buildings, vehicles, and vegetation. The resolutions of the images vary from $353 \times 278$ pixels for the smallest image to $4,294 \times 4,000$ pixels for the largest one. Consequently, the same pre-processing as the one described for the NucleiSeg dataset is applied to initial images and masks to generate input images and masks of $448 \times 448$ pixels, 30 times per initial pair of image and mask. In a similar fashion to other

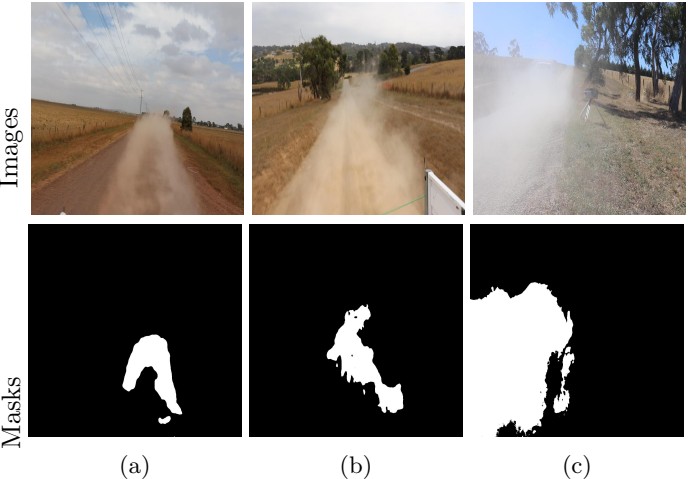

Figure 4: Example of images with their segmentation mask from the URDE dataset. One can notice that the masks have a cloudy, irregular structure.

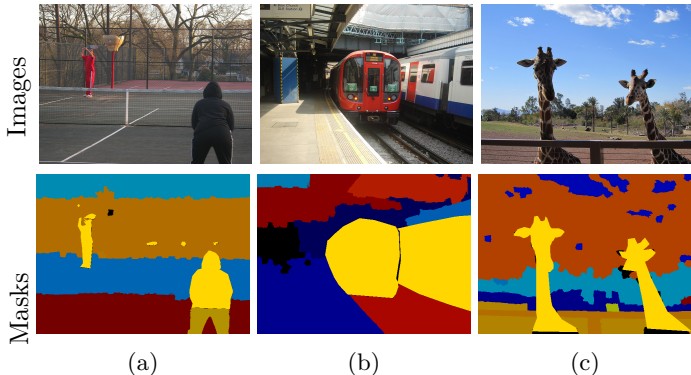

Figure 5: Example of images with their segmentation mask from the COCO-Stuff dataset. Those images illustrate the fact that orientation may matter. For instance, tennis players are expected to be standing up.

datasets, a few examples of paired images and masks contained in the dataset are presented in Figure 6. For this last dataset, the global orientation of the images is inherently arbitrary and does not carry meaningful information. To illustrate this, one can inspect Figure 6 (b), where the cars (dark blue segmentation masks) appear in different orientations across the image. Indeed, cars at the top of the image have a given orientation, while cars on the bottom-left have a different orientation. The variability in orientation for the same kind of objects is natural for aerial imagery, where objects photographed from above can appear rotated at any angle, but they are in fact the same object. This is exactly the idea captured by the property of rotation equivariance.

Table 1: Dataset Characteristics

| Dataset | N. images | Resolution (pixels) | Use Case | Orientation Sensitive |
|---|---|---|---|---|
| Kvasir-SEG | $1,000$ | $332 \times 487$ to $1,920 \times 1,072$ | Polyp segmentation. | ✗ |
| NucleiSeg | $135$ | $2,000 \times 2,000$ | Cell nuclei segmentation. | ✗ |
| URDE | $7,000$ | $1,024 \times 1,024$ | Roadside cloud dust segmentation. | ✓ |
| COCO-Stuff | $164,000$ | $59 \times 72$ to $640 \times 640$ | General object segmentation. | ✓ |
| iSAID | $2,806$ | $353 \times 278$ to $4,294 \times 4,000$ | Segmentation in satellite imagery. | ✗ |

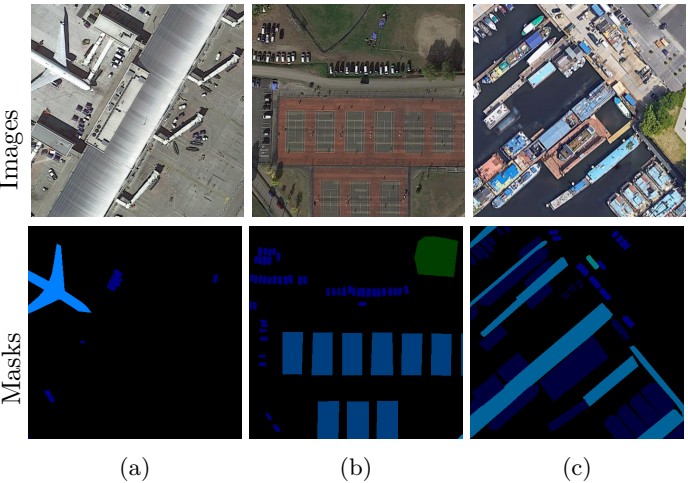

Figure 6: Example of images with their segmentation mask from the iSAID dataset. The orientation in those images does not matter as they are taken from an aerial perspective.

## 4.2  Model Architectures

After introducing the considered datasets, this section continues with a presentation of U-Net models that are used in this work and the motivation for their selection. These models can be divided in two main categories: (i) vanilla U-Nets and (ii) equivariant U-Nets. Within each category, we evaluate two model variants - a larger version and a smaller version - which differ in the number of trainable parameters. Furthermore, several ways of introducing the equivariance into U-Net are considered. To ensure a fair comparison, all models (and all variations of those models) share the same overarching architecture, with differences only in the convolutional approach, and in the number of filters in each layer (in order to compare models with the same number of trainable parameters). The following gives more details about the two categories of U-Net models.

**Vanilla U-Net**   The vanilla U-Net serves as the baseline model in this study. U-Net by Ronneberger et al. (2015) is a well-established Convolutional Neural Network (CNN) architecture, commonly used for image segmentation tasks (more details are presented in Section 2). This network relies on standard convolutional layers that are not inherently equivariant to rotations. As a result, this model serves as a baseline to measure the potential benefits of including rotation-equivariant mechanisms in alternative models. As indicated above, a small and a large version of this vanilla U-Net are developed.

The architecture of the vanilla U-Net is close to the one from the original paper by Ronneberger et al. (2015). It is made of two convolutional layers, followed by four downsampling (downsampling is achieved through max pooling layers that divide the resolution by a factor of 2) and four upscaling blocks (with bilinear upsampling that increases back the resolution by a factor 2, followed by convolution layers). Note that the use of a bilinear upsampling followed by a convolution is not the most common choice. Most of the time, transposed convolutions (or up-convolutions) are considered instead. However, those layers are not available in the case of equivariant models. Consequently, to ensure a fair comparison, transposed convolutions are replaced by this two steps process of an upsampling followed by a simple convolution. In practice, replacing the transposed convolutions with this process does not significantly affect the performance and involves the same number of trainable parameters.

All the convolutions in the model use a kernel size of $3 \times 3$, and are followed by batch normalization layers and ReLU (Nair & Hinton, 2010) activation functions. The first convolutional layer takes a fixed $224 \times 224$ input image. Finally, the last convolutional layer reduces the number of channels to the number of classes that are to be predicted.

All the elements regarding the U-Net configuration that are described in the previous paragraph are common for the two vanilla U-Net models (small and large), but they count a different number of trainable parameters. The small model contains roughly $500,000$ trainable parameters, while the large one contains roughly $11,000,000$ of them. In order to create differences in model size, a factor $\lambda \in \mathbb{R}$ is introduced to divide the number of filters in each layer of the model, thereby controlling the number of trainable parameters in the model. $\lambda$ has a constant, specific value for each model and size combination. For example, $\lambda = 1.65$ for the large vanilla U-Net, and $6.00$ for its small counterpart. The final number of filters for each convolutional layer is found in Table 2.

Table 2: Number of filters for the different models, for the two first convolutional layers (conv1, conv2), the four downsampling blocks (D1, D2, D3, D4, located on the left-hand side of Figure 1), and upsampling blocks (U1, U2, U3, U4, presented on the right-hand side of Figure 1).

| | Small Models | | | | | | | | | | Large Models | | | | | | | | | |
|---|---|---|---|---|---|---|---|---|---|---|---|---|---|---|---|---|---|---|---|---|
| | conv1 | conv2 | D1 | D2 | D3 | D4 | U1 | U2 | U3 | U4 | conv1 | conv2 | D1 | D2 | D3 | D4 | U1 | U2 | U3 | U4 |
| **vanilla** | 12 | 12 | 22 | 44 | 86 | 172 | 86 | 44 | 22 | 12 | 52 | 52 | 104 | 206 | 410 | 820 | 410 | 206 | 104 | 52 |
| **C4** | 4 | 4 | 6 | 12 | 24 | 46 | 24 | 12 | 6 | 4 | 14 | 14 | 28 | 56 | 112 | 222 | 112 | 56 | 28 | 14 |
| **C8/D4** | 4 | 4 | 6 | 10 | 18 | 34 | 18 | 10 | 6 | 4 | 10 | 10 | 20 | 40 | 78 | 154 | 78 | 40 | 20 | 10 |

**$E(2)$-U-Net**    The $E(2)$-U-Net modifies the vanilla U-Net architecture by incorporating equivariant convolutional blocks. These equivariant convolutional blocks follow the exact same architecture as for the vanilla U-Net, except that the group convolution implemented in $E(2)$-CNNs (Weiler & Cesa, 2019) is used. In addition, modified versions of ReLU activation functions and batch-normalization layers are also used. This is necessary given the fact that $E(2)$-CNNs involve a particular representation of the data that requires using specific activation functions and normalization. Furthermore, a final layer is also added at the end of the decoder part to perform the symmetry pooling over the different transformations in the considered symmetry group (see Section 2 for more details). Finally, an important modification is also the use of bigger filter sizes compared to vanilla convolutions. Indeed, many previous work on equivariant CNNs (Weiler & Cesa, 2019; Worrall et al., 2017; Delchevalerie et al., 2023) shows that using larger filter sizes is required to be able to achieve rotation equivariance (due to the use of a cartesian grid to represent images). This results in $E(2)$-U-Nets using a kernel size of $9 \times 9$ for convolutions, instead of $3 \times 3$ for the vanilla counterpart.

As with the vanilla U-Net, small and large versions of the models are defined by modifying the number of filters in each layer. Additionally, several symmetry groups are considered per $E(2)$-U-Net model:

- $C_4$**:** the cyclic group of order 4, which ensures equivariance to rotations of $90°$.

- $C_8$**:** the cyclic group of order 8, which extends equivariance to rotations of $45°$.

- $D_4$**:** the dihedral group of order 4, which combines $90°$ rotations and reflections.

This setup results in the creation of six $E(2)$-U-Net architectures (three small, and three large models). All the configurations are presented in Table 2, together with the number of filters in each layer.

## 5    Experiments

This section deals with the experiments performed in this study. First, the training setup is outlined, with details on the selected optimizer, loss function, and their hyperparameters. Second, details about the evaluation metrics for each task are presented. Third, the results obtained by the vanilla U-Net as well as the different $E(2)$-U-Net models across the different tasks are presented.

### 5.1    Training Setup

All models are trained under identical conditions to ensure a fair comparison. The selection of hyperparameters is based on the initial paper of each dataset, except for NucleiSeg, where the author's implementation deviated too much from approaches used in the four other datasets. Specifically, the authors use a batch size

of 128 of patched images having a size of $32 \times 32$ pixels and train their model over $600,000$ steps (Janowczyk & Madabhushi, 2016). Because our model accepts images with input size of 224, the resolution chosen by the authors is too small for our study. As a consequence, standard hyperparameters are retained for this dataset. The AdamW (Loshchilov & Hutter, 2019) optimizer is selected for all datasets, with a learning rate and a number of epochs that is specific to each dataset, as presented in Table 3. The Dice Loss (Sudre et al., 2017) is retained as the only loss criterion, because it is well-suited for segmentation tasks, in particular with imbalanced data. Similarly, the batch size is specific to each dataset and is also indicated in Table 3. Finally, the following data augmentation techniques are applied to images and their corresponding mask during training: random rotations of $0°, 90°, 180°$ or $270°$ with probability 0.25 each, horizontal and/or vertical flip(s) with probability 0.5 for each, and also a color normalization.

Table 3: Hyperparameters for Each Dataset

| Dataset | Batch Size | Learning Rate | N. epochs |
|---|---|---|---|
| Kvasir-SEG | 8 | $1 \times 10^{-4}$ | 150 |
| NucleiSeg | 16 | $1 \times 10^{-4}$ | 200 |
| COCO-Stuff | 16 | $1 \times 10^{-3}$ | 200 |
| URDE | 4 | $5 \times 10^{-4}$ | 500 |
| iSAID | 16 | $1 \times 10^{-4}$ | 250 |

For each model configuration, five-fold cross-validation is used. To do so, the dataset is partitioned into five non-overlapping subsets. For each of the five folds, four subsets are used for training while the remaining one is used for testing. This approach guarantees that each sample appears in the test set exactly once, and provides statistical robustness of the results. As already presented in Section 4, two different quantities of data are considered for training for all the different datasets. These two settings are defined by (i) "Large data setting" and (ii) "Small data setting" and correspond to the use of the full 4 folds for training for (i) , and the use of only a subset of 10% of each fold for (ii). Thus, (i) corresponds to the use of 80% of the available data for training, while (ii) corresponds to the use of only 8% of the initial dataset (i.e., 10% of the 80% of total data). For both settings, the testing set is held constant, and is the entirety of the remaining fold (made of 20% of the data). All models are trained using one Nvidia Tesla A100 with 40GB.

## 5.2 Evaluation metrics

The performance of the models is evaluated using metrics that are specific to the task at hand: binary segmentation or semantic segmentation.

For the binary segmentation tasks, one must classify each individual pixel as belonging either to the foreground class or to the background class. Five performance metrics are selected to evaluate the ability of a model to correctly identify the foreground class in images.

- **Dice Score (DS)** is a measure of the similarity (or overlap) between two sets, i.e., predicted segmentation masks (P) and ground truth masks (G). DS is computed as the ratio of twice the intersection of the predicted and ground truth masks to the sum of their total pixel counts. A perfect overlap has a DS of 1 since all pixels are forecasted in the correct class, while a DS of 0 means that there is absolutely no overlap between forecasted masks and their ground truth.

$$DS = \frac{2 \times |P \cap G|}{|P| + |G|}$$

- **Intersection over Union (IoU)** measures the overlap between the predicted segmentation masks (P) and their ground truth (G). IoU is defined as the intersection of the predicted and ground truth segments divided by their union.

$$IoU = \frac{|P \cap G|}{|P \cup G|}$$

- **Precision** is the ratio of True Positives (TP) to the sum of TP and False Positives (FP). This metric indicates the ability of a model to avoid false positives: the greater the ratio, the lower the relative FP.

$$\text{Precision} = \frac{TP}{TP + FP}$$

- **Recall** is the ratio of TP to the sum of TP and False Negatives (FN). Recall measures the ability of a model to detect the foreground class.

$$\text{Recall} = \frac{TP}{TP + FN}$$

- **Accuracy** is the proportion of correctly classified pixels, i.e., TP and True Negatives (TN) out of the total number of pixels.

$$\text{Accuracy} = \frac{TP + TN}{TP + TN + FP + FN}$$

In order to assess the performance of the models on the semantic segmentation tasks, four widely used evaluation criteria are selected (Caesar et al., 2018; Long et al., 2015; Eigen & Fergus, 2015; Caesar et al., 2016): (i) Mean Intersection over Union (mIoU) divides how many pixels are in the intersection between predictions and ground-truth class, and average the number over classes, (ii) Frequency-weighted IoU (FW IoU) is the per-class IoU, weighted by the frequency (at the pixel-level) of each class, (iii) Mean Accuracy (mAcc), is the average pixel accuracy calculated across all classes, and (iv) Pixel Accuracy (pAcc) is the percentage of pixels that are correctly labeled. Note that these criteria are derived from binary segmentation metrics described above.

## 5.3  Results

This section presents the results obtained for the five datasets. For each of them, the results for the small and large models, both for the vanilla and $E(2)$-U-Net are presented. Each configuration has two settings: a small data setting (i.e., only 10% of each training fold is used), and a large data setting (i.e., the full 4 folds are used for training). This means that for each dataset, each of the four models (one vanilla, and three $E(2)$-U-Net) has four variations: small/large number of trainable parameters (two variations), and small/large data setting (two additional variations). This resulted in the training of 16 models, for each of the five datasets.

**Kvasir-SEG**    Results for the Kvasir-SEG dataset are summarized in Table 4. In the large data setting, and regarding the small models, the $E(2)$-U-Net model based on the use of the $C_8$ symmetry group outperforms the others in terms of Dice Score, IoU, precision, and accuracy, whereas the vanilla U-Net achieves the highest recall. Regarding the larger models for the same data setting, the $E(2)$-U-Net based on $C_8$ again shows the best performance in Dice Score, IoU , accuracy, and recall. With regards to precision, both $C_8$ and $D_4$ show comparable performance. In the small data setting, and regarding small models, $C_8$ has the highest Dice Score, IoU, precision, and accuracy while $C_4$ has the highest recall. As far as the large models and small data setting is concerned, the overall picture is comparable to that of the small models for the same data setup in terms of DS and IoU, where $C_8$ has the highest metrics. Interestingly, the vanilla model has the best recall for the large model. Overall, the equivariant models show better performance than the vanilla counterparts, both regarding small and large models.

While the above analysis helps in understanding models' performance when there is no constraint on training resources, a second analysis is performed to assess the performance from a sustainability standpoint that considers the resources used. To do so, one can visualize the evolution of performance against the cumulative time during training. Therefore, we also present in Figure 7 the IoU with respect to the cumulative training time (in seconds) for the large as well as small models, both for the small and large data setting case. Please note that the total number of epochs is the same for all models and all configurations. Under the large data setup in this Figure, one sees the vanilla U-Net is better than the equivariant ones until it stops training at $3 \times 10^2$ and $7 \times 10^2$ seconds, for small and large models respectively. Thus, if one had to stop training early,

Table 4: Performance metrics for the Kvasir-SEG dataset

| Model | Large data setting | | | | | Small data setting | | | | |
|---|---|---|---|---|---|---|---|---|---|---|
| | Dice | IoU | Pre. | Rec. | Acc. | Dice | IoU | Pre. | Rec. | Acc. |
| **Small models** | | | | | | | | | | |
| *vanilla* | 80.5 ±2.5 | 68.0 ±3.1 | 79.2 ±6.1 | **83.5** ±4.4 | 93.9 ±1.2 | 54.4 ±6.7 | 38.0 ±6.0 | 65.4 ±13.3 | 52.6 ±11.1 | 87.2 ±2.8 |
| $C_4$ | 80.6 ±3.2 | 68.3 ±4.3 | 84.2 ±3.8 | 79.2 ±4.4 | 94.2 ±1.1 | 50.5 ±9.5 | 35.3 ±7.9 | 55.0 ±14.7 | **63.1** ±14.0 | 72.9 ±20.2 |
| $C_8$ | **83.1** ±2.9 | **71.8** ±4.0 | **86.7** ±1.5 | 81.2 ±5.3 | **95.0** ±0.9 | **60.4** ±2.6 | **44.1** ±2.6 | **74.9** ±8.3 | 53.0 ±5.0 | **89.6** ±0.5 |
| $D_4$ | 81.1 ±3.6 | 69.0 ±4.8 | 85.0 ±4.5 | 79.0 ±5.0 | 94.4 ±1.3 | 49.8 ±9.2 | 34.7 ±8.0 | 57.6 ±14.9 | 59.0 ±15.4 | 74.0 ±20.0 |
| **Large models** | | | | | | | | | | |
| *vanilla* | 86.5 ±1.6 | 76.7 ±2.3 | 89.5 ±1.1 | 84.7 ±2.7 | 96.0 ±0.5 | 66.5 ±3.8 | 50.8 ±4.1 | 76.2 ±5.8 | **61.3** ±9.9 | 90.7 ±0.4 |
| $C_4$ | 87.2 ±1.2 | 77.8 ±1.7 | 88.8 ±2.0 | 86.6 ±1.7 | 96.2 ±0.4 | 66.7 ±3.3 | 51.1 ±3.5 | **83.9** ±5.0 | 57.4 ±3.7 | 91.4 ±0.5 |
| $C_8$ | **88.0** ±1.9 | **79.0** ±2.8 | **90.2** ±1.2 | **86.8** ±3.4 | **96.4** ±0.7 | **68.3** ±3.9 | **53.0** ±4.7 | 82.7 ±2.7 | 60.0 ±4.8 | **91.7** ±0.5 |
| $D_4$ | 86.4 ±1.9 | 76.7 ±3.1 | **90.2** ±1.6 | 84.0 ±2.7 | 96.0 ±0.7 | 63.8 ±2.7 | 47.7 ±2.8 | 83.5 ±7.4 | 53.4 ±4.5 | 90.9 ±0.6 |

the vanilla model would be better, leading to a gain in computational resources. The same observation can be made about the small data setup.

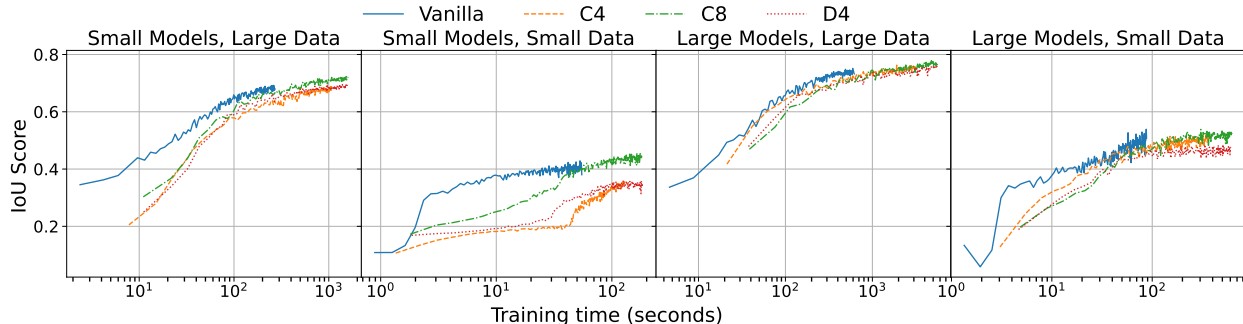

Figure 7: Kvasir-SEG: IoU performance over cumulative time (in seconds) for small and large models in small and large data setting, averaged across the 5 folds. The total number of epochs is the same for all models and all configurations.

**NucleiSeg** Results for the NucleiSeg dataset are summarized in Table 5. Regarding the large data setting for the small models, one can observe that the vanilla U-Net model has the highest Dice Score, IoU and recall. Considering the precision metric, the $E(2)$-U-Net based on the $C_8$ symmetry group achieves the best score. As far as the accuracy is concerned, $C_8$ and $D_4$ both show the highest score. For the large data setting and the large models, the vanilla U-Net model has the best Dice Score, IoU, precision, and recall. $C_4$ has the highest accuracy level. The small data setting reveals that vanilla models for both small and large models have the best Dice Score and IoU. The other metrics are dominated by $C_8$ or $D_4$. Overall, the vanilla models seem to perform better than the equivariant counterparts for this dataset.

Figure 8 presents the IoU with respect to the cumulative time in seconds (with a constant number of epochs throughout models and setups). It shows that vanilla models have the greatest performance throughout. Also, one can notice that the performance starts dropping around time $2 \times 10^2$ and $5 \times 10^2$ seconds for small and large models respectively, under the large data setup. This is also true for the small data setup, with a drop of performance at $1 \times 10^2$ and $1 \times 10^2$ seconds.

**URDE** Results for the URDE dataset are summarized in Table 6. Focusing on the large data setting and small models, the $E(2)$-U-Net based on $C_4$ and $D_4$ models achieves comparable Dice and IoU. The $D_4$ variant also has the highest precision, while the $C_8$ model outperforms the other models in recall. The equivariant configurations all achieve highest accuracy metrics. For the larger models under the large data setting, the $E(2)$-U-Net based on $C_4$ shows the best Dice Score, IoU, precision and recall. It also achieves the highest accuracy, a score matched by $C_8$. The small data setting shows that the best large model is $C_4$

Table 5: Performance metrics for the NucleiSeg dataset

| Model | Large data setting | | | | | Small data setting | | | | |
|---|---|---|---|---|---|---|---|---|---|---|
| | Dice | IoU | Pre. | Rec. | Acc. | Dice | IoU | Pre. | Rec. | Acc. |
| **Small models** | | | | | | | | | | |
| *vanilla* | **28.4** ±7.6 | **17.3** ±5.1 | 37.4 ±3.6 | **30.1** ±13.8 | 97.0 ±0.7 | **18.7** ±7.3 | **10.8** ±4.6 | 25.2 ±3.2 | 20.5 ±10.5 | 96.6 ±0.4 |
| $C_4$ | 25.7 ±2.8 | 15.2 ±1.8 | 33.3 ±2.8 | 25.0 ±5.5 | 97.0 ±0.2 | 15.1 ±3.6 | 8.4 ±2.0 | 27.3 ±2.7 | 12.1 ±3.1 | 97.2 ±0.3 |
| $C_8$ | 23.4 ±4.3 | 13.8 ±2.8 | **37.8** ±3.4 | 19.8 ±5.6 | **97.3** ±0.3 | 13.8 ±6.2 | 7.7 ±3.7 | **30.5** ±4.2 | 10.4 ±6.0 | **97.4** ±0.4 |
| $D_4$ | 24.5 ±5.2 | 14.5 ±3.3 | 37.6 ±4.3 | 21.2 ±6.9 | **97.3** ±0.2 | 12.9 ±5.9 | 7.1 ±3.5 | 22.8 ±14.5 | **29.2** ±9.5 | 78.3 ±2.7 |
| **Large models** | | | | | | | | | | |
| *vanilla* | **39.8** ±2.3 | **25.9** ±1.7 | **36.9** ±1.7 | **53.3** ±4.7 | 96.5 ±0.6 | **24.9** ±8.3 | **15.1** ±5.5 | 36.1 ±7.1 | **26.0** ±16.9 | 97.0 ±0.8 |
| $C_4$ | 34.4 ±1.7 | 21.6 ±1.2 | 34.8 ±2.3 | 40.1 ±5.6 | **96.8** ±0.5 | 15.8 ±3.6 | 9.0 ±2.1 | **38.4** ±1.6 | 11.4 ±3.1 | **97.5** ±0.4 |
| $C_8$ | 34.8 ±2.3 | 21.8 ±1.9 | 34.4 ±3.2 | 42.4 ±3.6 | 96.7 ±0.4 | 14.4 ±4.1 | 8.2 ±2.5 | 37.4 ±4.0 | 10.4 ±4.4 | **97.5** ±0.4 |
| $D_4$ | 33.7 ±3.1 | 21.0 ±2.4 | 33.3 ±2.6 | 43.5 ±6.8 | 96.4 ±0.6 | 13.0 ±5.2 | 7.3 ±3.1 | 37.2 ±3.3 | 9.2 ±4.4 | **97.5** ±0.5 |

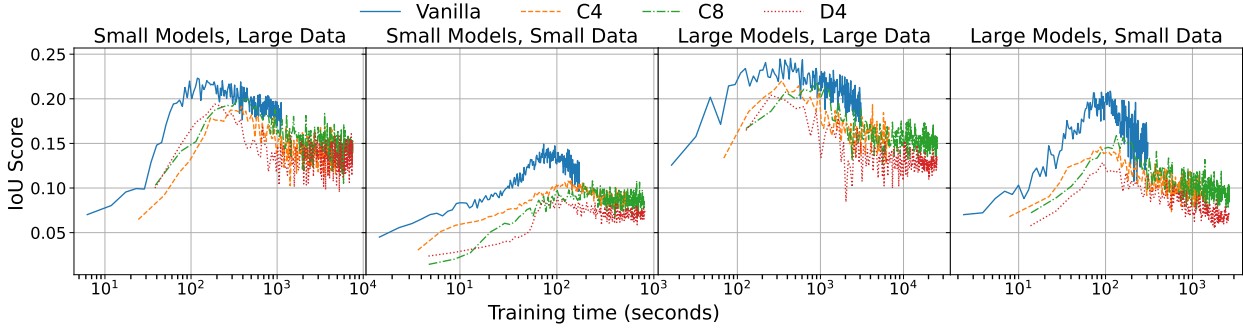

Figure 8: NucleiSeg: IoU performance over cumulative time (in seconds) for small and large models in small and large data setting, averaged across the 5 folds. The total number of epochs is the same for all models and all configurations.

for all performance metrics. Regarding the small models of this same setting, $C_4$ has the higest Dice Score, IoU, recall and accuracy, whereas $C_8$ has the best precision.

Table 6: Performance metrics for the URDE dataset

| Model | Large data setting | | | | | Small data setting | | | | |
|---|---|---|---|---|---|---|---|---|---|---|
| | Dice | IoU | Pre. | Rec. | Acc. | Dice | IoU | Pre. | Rec. | Acc. |
| **Small models** | | | | | | | | | | |
| *vanilla* | 78.1 ±2.4 | 70.9 ±2.2 | 41.4 ±3.1 | 39.3 ±3.9 | 98.5 ±0.1 | 65.9 ±4.3 | 58.6 ±4.7 | 31.7 ±3.5 | 33.2 ±4.5 | 97.9 ±0.2 |
| $C_4$ | 82.0 ±1.6 | **75.0** ±1.7 | 43.5 ±2.7 | 42.1 ±2.4 | **98.7** ±0.1 | **67.9** ±2.6 | **60.4** ±2.6 | 37.5 ±2.7 | **33.4** ±2.3 | **98.2** ±0.1 |
| $C_8$ | 81.6 ±2.1 | 74.7 ±2.2 | 43.5 ±2.1 | **42.3** ±2.8 | 98.7 ±0.1 | 66.4 ±4.9 | 58.7 ±4.8 | **37.8** ±2.5 | 32.9 ±2.0 | 98.1 ±0.1 |
| $D_4$ | **82.1** ±1.3 | **75.0** ±1.4 | **44.0** ±2.2 | 41.9 ±2.7 | 98.7 ±0.1 | 65.9 ±2.9 | 58.1 ±2.9 | 37.5 ±3.0 | 32.8 ±2.6 | 98.1 ±0.1 |
| **Large models** | | | | | | | | | | |
| *vanilla* | 81.3 ±1.0 | 74.1 ±0.8 | 42.2 ±1.6 | 40.4 ±2.3 | 98.6 ±0.1 | 62.1 ±8.4 | 54.8 ±8.8 | 32.4 ±1.9 | 32.0 ±3.5 | 97.7 ±0.5 |
| $C_4$ | **84.4** ±1.5 | **77.6** ±1.4 | **44.6** ±1.9 | 43.0 ±3.3 | **98.8** ±0.1 | **71.3** ±2.2 | **63.5** ±1.9 | **39.1** ±3.2 | **33.4** ±2.9 | **98.2** ±0.1 |
| $C_8$ | 83.6 ±1.6 | 76.6 ±1.6 | 43.2 ±1.8 | **43.1** ±3.3 | **98.7** ±0.1 | 70.6 ±2.9 | 63.0 ±2.9 | 38.0 ±1.7 | 33.1 ±1.5 | **98.2** ±0.1 |
| $D_4$ | 84.2 ±1.5 | 77.2 ±1.4 | 44.5 ±2.2 | 42.8 ±2.7 | **98.7** ±0.1 | 71.0 ±1.9 | 63.2 ±1.9 | 38.6 ±2.5 | 32.9 ±1.2 | 98.1 ±0.0 |

Regarding the time-performance plot (with the same total number of epochs for all configurations and models) presented in Figure 9, it is clear that for the small data setting as well as for the large data setting, the vanilla U-Net performs significantly worse than the three equivariant models in terms of IoU, both for the small and large models.

**COCO stuff** Results for the COCO-Stuff dataset are presented in Table 7. For small models in the large data setting, the vanilla U-Net shows better performance than the $E(2)$-U-Nets for all metrics, except for

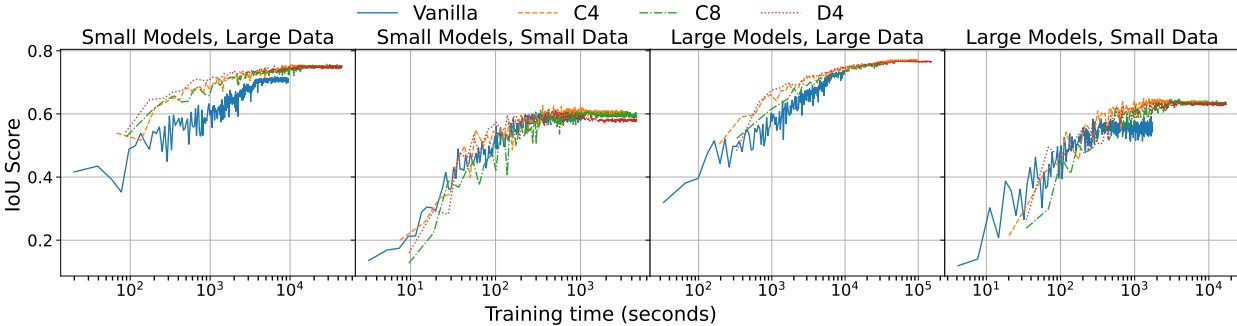

Figure 9: URDE: IoU performance over cumulative time (in seconds) for small and large models in small and large data setting, averaged across the 5 folds. The total number of epochs is the same for all models and all configurations.

mean accuracy, where its performance is equal to that of $C_4$. However, when considering larger models, $C_4$ and $C_8$ models close the gap in performance and $C_8$ even achieves the best mean IoU and mean accuracy. The $D_4$ model consistently underperforms. In the small data setting, however, the vanilla model is best across all metrics, both for small and large models.

Table 7: Performance metrics for the COCO-Stuff dataset

| | Large data setting | | | | Small data setting | | | |
|---|---|---|---|---|---|---|---|---|
| **Model** | Mean IoU | Pixel Acc. | Mean Acc. | fW IoU | Mean IoU | Pixel Acc. | Mean Acc. | fW IoU |
| **Small models** | | | | | | | | |
| *vanilla* | **13.6** ±0.2 | **64.2** ±0.3 | **18.9** ±0.2 | **49.1** ±0.5 | **10.8** ±0.2 | **60.1** ±0.3 | **15.7** ±0.3 | **44.6** ±0.6 |
| $C_4$ | 13.5 ±0.1 | 64.0 ±0.3 | **18.9** ±0.2 | 48.9 ±0.5 | 9.5 ±0.2 | 56.9 ±0.4 | 14.1 ±0.2 | 41.4 ±0.4 |
| $C_8$ | 13.0 ±0.6 | 63.0 ±1.0 | 18.4 ±0.9 | 47.7 ±1.4 | 9.6 ±0.3 | 57.6 ±0.4 | 14.3 ±0.3 | 42.6 ±0.3 |
| $D_4$ | 12.9 ±0.5 | 62.9 ±0.7 | 18.2 ±0.6 | 47.4 ±0.9 | 8.6 ±0.3 | 55.1 ±0.7 | 12.7 ±0.3 | 40.9 ±0.6 |
| **Large models** | | | | | | | | |
| *vanilla* | 17.9 ±0.1 | **68.9** ±0.2 | 23.8 ±0.3 | **56.3** ±0.6 | **10.9** ±0.3 | **60.9** ±0.4 | **15.2** ±0.4 | **47.7** ±0.5 |
| $C_4$ | 18.2 ±0.2 | 67.8 ±0.2 | 24.1 ±0.2 | 54.8 ±0.3 | 10.6 ±0.3 | 60.4 ±0.4 | 14.9 ±0.3 | 47.2 ±0.4 |
| $C_8$ | **18.3** ±0.4 | 67.8 ±0.2 | **24.2** ±0.6 | 54.8 ±0.7 | 10.4 ±0.1 | 60.3 ±0.3 | 14.5 ±0.1 | 47.2 ±0.1 |
| $D_4$ | 16.7 ±0.3 | 65.9 ±0.5 | 22.5 ±0.2 | 53.4 ±0.1 | 9.1 ±0.2 | 58.7 ±0.4 | 12.9 ±0.3 | 45.6 ±0.5 |

The analysis of Figure 10 (similarly to the previous analyses, total number of epochs is constant) shows that under the large data setting, the small vanilla model is always better than its equivariant counterparts in terms of mean IoU. However, when looking at large models, it is not always the case. Indeed, between times $1.5 \times 10^3$ and $8 \times 10^4$, one notices that very similar performance between vanilla and $C_4$ models. Ultimately, $C_4$ and $C_8$ finish at higher mean IoU levels than the vanilla U-Net. For the small data setting, the vanilla model is always the best in terms of mean IoU.

**iSAID**  Results for the iSAID dataset are presented in Table 8. As far as both the small and large data settings are concerned, and for both the small and large models, the vanilla U-Net outperforms the three other models across all metrics, with a clear advantage in mean IoU and mean accuracy.

The analysis of mean IoU against cumulative time (in seconds) in Figure 11 shows that the vanilla models are always the best, for the small and the large data setup, both for small and large models.

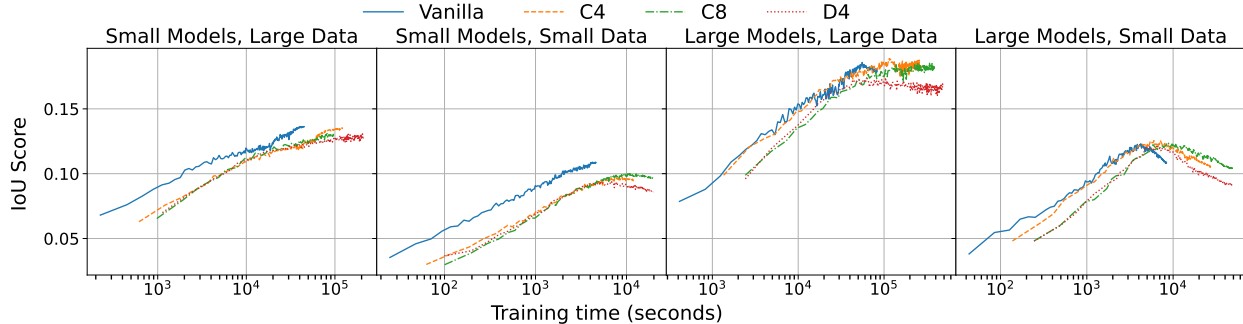

Figure 10: COCO-Stuff: mean IoU performance over cumulative time (in seconds) for small and large models in small and large data setting, averaged across the 5 folds. The total number of epochs is the same for all models and all configurations.

Table 8: Performance metrics for the iSAID dataset

| | Large data setting | | | | Small data setting | | | |
|---|---|---|---|---|---|---|---|---|
| **Model** | Mean IoU | Pixel Acc. | Mean Acc. | fW IoU | Mean IoU | Pixel Acc. | Mean Acc. | fW IoU |
| **Small models** | | | | | | | | |
| *vanilla* | **39.9** ±1.6 | **97.2** ±0.3 | **43.5** ±1.7 | **95.1** ±0.4 | **30.3** ±1.7 | **95.9** ±0.5 | **33.3** ±1.8 | **93.1** ±0.8 |
| $C_4$ | 33.4 ±2.6 | 96.8 ±0.3 | 36.7 ±2.8 | 94.5 ±0.4 | 23.5 ±1.2 | 95.1 ±0.4 | 25.8 ±1.2 | 91.9 ±0.7 |
| $C_8$ | 34.9 ±2.0 | 97.1 ±0.2 | 38.6 ±1.9 | 94.9 ±0.4 | 24.6 ±2.0 | 95.6 ±0.3 | 27.1 ±1.9 | 92.6 ±0.5 |
| $D_4$ | 30.8 ±2.9 | 96.8 ±0.3 | 34.1 ±3.2 | 94.4 ±0.5 | 20.3 ±1.2 | 95.0 ±0.4 | 22.4 ±1.3 | 91.7 ±0.7 |
| **Large models** | | | | | | | | |
| *vanilla* | **48.2** ±1.8 | **97.6** ±0.3 | **52.7** ±1.7 | **95.8** ±0.4 | **36.2** ±2.0 | **96.9** ±0.3 | **39.7** ±2.0 | **94.6** ±0.4 |
| $C_4$ | 43.1 ±1.6 | 97.5 ±0.2 | 47.1 ±1.7 | 95.5 ±0.4 | 32.1 ±1.0 | 96.3 ±0.3 | 35.0 ±1.0 | 93.6 ±0.5 |
| $C_8$ | 44.6 ±1.4 | 97.6 ±0.3 | 48.8 ±1.6 | 95.7 ±0.4 | 32.3 ±1.0 | 96.5 ±0.3 | 35.2 ±1.0 | 93.9 ±0.5 |
| $D_4$ | 40.4 ±0.2 | 97.3 ±0.3 | 44.2 ±0.2 | 95.3 ±0.4 | 26.3 ±0.6 | 96.1 ±0.3 | 28.7 ±0.8 | 93.2 ±0.5 |

# 6 Discussion

The analysis of the results reveals interesting patterns regarding the usefulness of equivariant models across model sizes, the type of tasks, the dataset specificity (i.e., the size, nature of images, and number of classes) and the data regime. This section sums up the observations from the previous section, and tries to extract

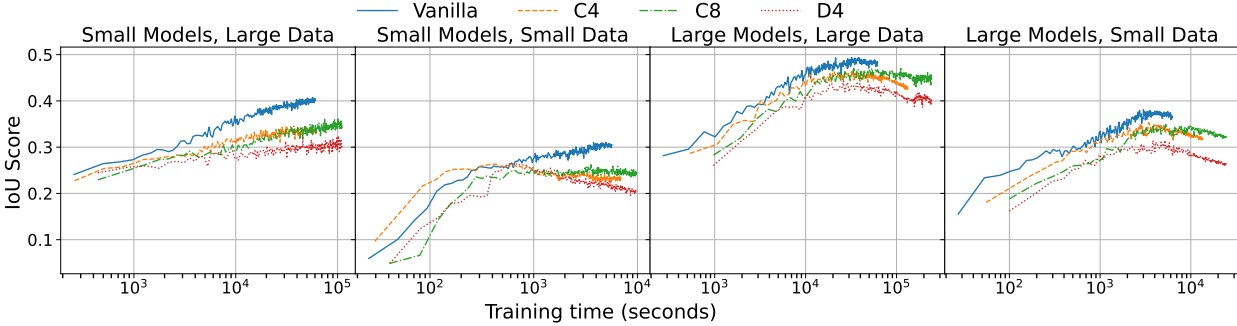

Figure 11: iSAID mean IoU performance over cumulative time (in seconds) for small and large models in small and large data setting, averaged across the 5 folds. The total number of epochs is the same for all models and all configurations.

some insights about when it could be interesting to consider equivariant U-Net for segmentation tasks. Based on the results and previous studies in the literature (see Section 2), the discussion is subdivided into two parts, depending on the interest one has in using equivariant models: (i) getting an improvement in terms of evaluation metrics ("are equivariant models better at solving the considered task?") and (ii) getting an improvement in terms of sustainability ("are equivariant models able to achieve comparable performance while reducing the amount of required computational resources?").

## 6.1 Discussion based on evaluation metrics

First, we present our key findings on evaluation metrics. We then discuss the overall performance independently for the different setups in terms of model size and data regime (small or large). In particular, we discuss performance for (i) small models in large data settings, (ii) large models in large data settings, (iii) small models in small data settings, and (iv) large models in small data settings.

**Key findings**  Our analysis reveals that the effectiveness of equivariant U-Nets compared to their vanilla counterparts largely depends on the nature of the objects as well as the complexity of the segmentation task. For datasets where objects can have arbitrary shapes, and their orientation is not semantically meaningful, such as Kvasir-SEG and URDE, we find that equivariant models often outperform vanilla architectures (see Table 4 and Table 6). However, when objects are symmetric from a rotational standpoint (e.g., nuclei have circular shapes in NucleiSeg), vanilla U-Nets show better performance than their rotation equivariant counter parts. One explanation is that the rotation equivariance constrains the filters and reduces their expressiveness, which hinders the learning process of the model. For general-purpose segmentation, like COCO-Stuff, equivariant U-Nets match the vanilla ones on performance metrics (see Table 7). For those general tasks, the performance gap arises in small data regimes, where equivariant models are consistently worse than the vanilla ones. This shows that low data availability affects the trade-off between architectural constraints and model expressiveness. In particular, $D_4$ consistently shows lower performance across datasets, suggesting that stronger symmetry constraints may be too restrictive for practical segmentation tasks.

**Small models in large data settings**  In terms of evaluation metrics, equivariant models can be preferable to their vanilla counterparts depending on the task. For example, it is the case for the Kvasir-SEG dataset and the URDE dataset, where results show that $E(2)$-U-Net can achieve better performance (only requiring slightly more computation time for Kvasir-SEG). However, as far as the NucleiSeg dataset is concerned, vanilla U-Nets seem to be a better choice than equivariant U-Nets, as they achieve better or at least top performance regarding all the different metrics. This can be explained by the fact that object of interest in NucleiSeg are all of circular shape. In such situation where the object of interest already satisfy the rotational symmetry by itself (rotation of the object of interest has no impact on the observed shape), applying constraints on the filters like in group-CNNs is meaningless, and even leads to an unnecessary reduction of the expressiveness of the filters, leading to lower levels of performance.

Regarding the COCO-Stuff and iSAID datasets, the vanilla U-Nets achieve better overall performance. However, for COCO-Stuff, the $C_4$ equivariant U-Net is able to achieve similar performance, only requiring a larger training time. Regarding the iSAID dataset, equivariant models seems to struggle achieving the vanilla performance, which is even more true for the $D_4$ equivariant model. This may be explained by the fact that the prior knowledge of rotation equivariance is not worth the reduction in expressiveness in the filter. Indeed, only rotation equivariant features can be extracted, which is a constraint that reduces the search space when optimizing the filters, and which is sometimes not beneficial. This insight can also be confirmed by looking at the $D_4$ models that are always significantly below the other ones in terms of performance, and which further constrain the shape of the filters by including a reflection symmetry.

**Large models in large data settings**  Regarding the first three datasets, i.e., Kvasir-SEG, NucleiSeg and URDE, conclusions are very similar to the previous ones using small models. However, one can point out that large models are always significantly better than their lightweight versions. Now, the results for the COCO-Stuff and iSAID datasets are more interesting. Indeed, for the first one, performance is significantly better, and the $C_4$ and $C_8$ models are now even better than the vanilla models according to mean IoU and

mean accuracy. For the iSAID dataset, vanilla U-Net are still the best performing ones and large differences still exist with their equivariant counterparts.

**Small models in small data settings**   The results for this experimental setup are really similar to those already discussed. However, one can point out that the gap in performance is sometimes bigger. It means that significantly reducing the amount of data seems to exacerbate the advantage and disadvantage of the different methods.

**Large models in small data settings**   Again, insights are similar to those already provided for the other settings. Using larger models seems beneficial as it significantly increases the overall performance. The URDE dataset is the only case where, for the vanilla U-Net, using a larger model leads to a small loss in performance. The observation of Figure 9 shows that the larger model under small data quickly stagnates at around 55% of IoU, with large up and down peaks in the metric between $5 \times 10^2$ and $3 \times 10^3$ seconds. The behaviour of the smaller model under the small data setting does not share this characteristic, and IoU remains at around 59% from $5 \times 10^2$ seconds until the end of training.

**Final thoughts**   Based on the results from Kvasir-SEG and NucleiSeg, one may conclude with a first insight that **equivariant U-Nets are an interesting choice to achieve better performance in tasks for which local rotations in the image have an impact on the final image, but does not contain any information**. It is the case for Kvasir-SEG, as polyps can be of arbitrary shape. However, the particular orientation of polyps in the image should not affect their segmentation. This last point is also true for NucleiSeg, but in this case **as nuclei are not of arbitrary shape but are instead always of circular shape, using rotation equivariance is detrimental to the performance**. One additional factor of explanation lies with the vast quantity of objects that are present in each image, making the task more difficult for equivariant U-Nets.

Regarding datasets that focus on more general tasks, such as COCO-Stuff, equivariant models can achieve similar performance to their vanilla counterparts. However, **if equivariant features are used alone (without introducing non-equivariant features in the model), results show that it is preferable to consider sufficiently large models**. Indeed, the loss in expressiveness in the filters makes it difficult for the small models to capture enough features in the images. Nonetheless, the fact that equivariant U-Nets are able to achieve similar performance when increasing the number of filters indicate that they are able to extract meaningful features. Thus, for tasks that are more general and involve complex object segmentations, **one could benefit from new models that are specifically designed to extract both equivariant and non-equivariant features in parallel**.

## 6.2   Discussion in terms of computational resources

Achieving the best performance in terms of evaluation metrics is of course of primary importance for many tasks. However, it is also important to consider the amount of computational resources (and time) that are involved in the training process of the models. For example, for some applications, it could be interesting to consider a particular model even if it leads to slightly lower metric performance (e.g. lower IoU) if it could save a significant amount of time. For this purpose, $E(2)$-U-Net are an interesting candidate. This second part of the discussion will focus more on the question of sustainability. In other words: "is it possible to obtain a good trade-off between performance and training time?"

For many of the experimental setups that were considered, $E(2)$-U-Nets are able to achieve better or similar performance. However, we only observe for a few particular cases (URDE, small and large models on large data settings, and iSAID, small models and small data settings) potential gains in terms of computational resources. In other cases, the IoU with respect to the training time are similar or convergence is slightly slower for equivariant models. This can be explained by the fact that even if rotation equivariance is meaningful, using group-convolutions is more expensive than vanilla convolutions. The analysis of the results reveals that adding more computational resources does not always justify the marginal improvements in epoch-wise convergence. This finding challenges the assumption that equivariant models are universally beneficial for reducing training time, as the optimal choice depends heavily on the specific task requirements. The experi-

ments consistently show that both larger models and larger data regimes correlate with higher performance. Nonetheless, **the relationship between these factors reveals an important asymmetry: reducing the training dataset size has a substantially more detrimental effect on model performance compared to reducing model size, despite both approaches decreasing training time**. If one wants to obtain a good trade-off between performance and training time, using smaller models appears to be a better approach over using smaller training dataset size.

## 7 Conclusion and limitations

In this work, the performance of vanilla and equivariant U-Nets is assessed across five diverse datasets, namely Kvasir-SEG, NucleiSeg, URDE, COCO-Stuff, and iSAID. They vary in size, may or may not have rotation-invariant characteristics, and focus on specific, narrow tasks or on tasks that are more general. This enables the examination of the effects of model size, dataset characteristics, and segmentation task type (i.e., binary or semantic segmentation). The intention behind such analysis is to get a better understanding of how architectural constraints, such as rotation equivariance, impact the performance and computational efficiency of segmentation models.

The findings of this study highlight that equivariant U-Nets (and more specifically, rotation equivariant models), particularly $E(2)$-U-Nets based on the use of discrete symmetry groups like $C_4$, $C_8$ or $D_4$, provide advantages in datasets with high orientation variability, such as Kvasir-SEG and URDE, where equivariant models outperformed their vanilla counterparts. However, in datasets like NucleiSeg, where the objects exhibit inherent rotational symmetry, vanilla U-Nets achieved higher accuracy and expressiveness. For larger, more complex datasets such as COCO-Stuff and iSAID, vanilla U-Nets consistently demonstrated superior performance. This shows that equivariance constraints may not always translate to improved performance, especially in settings requiring broader feature expressiveness. Moreover, we find that reducing the number of training data (i.e., small data setting with only 10% of each training fold data) does not have a significant impact on the relative performance.

An important future work will be to design and test models that are able to build a set of both equivariant and non-equivariant features. Our experiments highlight the fact that for more complex datasets like COCO-Stuff or iSAID, equivariance is less useful. Still, they are able to extract useful features and can achieve almost state-of-the-art performance. Thus, we believe that those equivariant features can extract complementary information that could benefit to the model.

### Acknowledgments

Valentin Delchevalerie is supported by SPWR under grant n°2010235 - ARIAC by DIGITALWALLONIA4.AI. The present research benefited from computational resources made available on Lucia, the Tier-1 supercomputer of the Walloon Region, infrastructure funded by the Walloon Region under the grant agreement n°1910247.

The authors thank Pragati Mitra and Sacha Corbugy for their comments and the fruitful discussions on this paper.

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
