# OpenReview forum: "On the effectiveness of Rotation-Equivariance in U-Net: A Benchmark for Image Segmentation"
_TMLR — Accepted by TMLR_

### Review · Reviewer_uWjN · 2025-03-04

**Summary Of Contributions:**

This paper presents a comprehensive benchmark study evaluating the effectiveness of rotation-equivariant U-Net architectures for image segmentation across a diverse set of datasets, including medical imaging, remote sensing, and natural scene segmentation. The authors systematically compare vanilla U-Net with several rotation-equivariant variants (e.g., group-equivariant U-Nets), assessing their performance under both large-data and small-data settings.

**Audience:**

Yes

**Broader Impact Concerns:**

1. The selected datasets, while diverse, may underrepresent segmentation challenges in low-resource regions, introducing bias that could steer future research away from culturally or geographically underrepresented segmentation tasks requiring different invariance properties.

2. The benchmark’s focus on controlled datasets may overlook deployment challenges, such as sensor noise, varying image quality, or domain shifts, limiting its applicability to real-world segmentation tasks in unpredictable environments.

**Claims And Evidence:**

Yes

**Requested Changes:**

Requested Changes

1. The author should add background and related works for image segmentation in section 2 and 3.

2. In section 4, the author should provide references to support the statement as I show in weakness 3.

3. The authors should ensure that their statements are consistent with the results presented in the tables to avoid confusion and improve the clarity of their analysis. See details in weakness 5.

1 and 3 are critical to securing my recommendation for acceptance. 2 would simply strengthen the work in my view.

**Strengths And Weaknesses:**

Strengths:

1. The paper provides a comprehensive benchmark evaluating rotation-equivariant U-Net variants across multiple segmentation datasets, offering broad insights into their effectiveness across medical, remote sensing, and general segmentation tasks with varying rotation invariance requirements.

2. By comparing performance in both large and small data settings, the paper highlights how rotation-equivariant models behave in low-data scenarios, providing practical guidance for segmentation tasks with limited annotated data.

Weaknesses:

1. The authors aim to build a benchmark for image segmentation tasks. However, in the background section, the focus is primarily on U-Net and equivariance in CNNs, with no introduction or context provided for image segmentation itself. Adding a brief overview of image segmentation as a task, its importance, and common challenges would help better position the benchmark within the broader research landscape.

2. In the first paragraph of related work, the author firstly shows several related works about classification tasks. However, there is no logic that can show there is a limited number of works in segmentation tasks. The author should show more related work in segmentation tasks.

3. In section 4, the author states “This is motivated by the idea that equivariant models are constrained by a prior knowledge, which may reduce their required number of parameters to obtain a model with similar performance” in the first paragraph, however, this statement lacks a citation to support it. To improve credibility, the authors should provide references to prior theoretical or empirical works that support this claim about equivariance reducing parameter requirements.

4. For section 4.1, the author states “Consequently, these five datasets cover a large spectrum of segmentation tasks: smaller/larger datasets, binary/semantic segmentation, and rotation invariant/standard applications”. However, before the sentence, the author only discusses the number of classes of different datasets and has no information about  invariant/standard applications.

5. In paragraph of Kvasir-SEG in section 5.3, the author states “In small data setting, the small models show better Dice Score, IoU, precision, and accuracy for C8 while C4 has the highest recall”. However, in table 4, it shows that in small data settings, large models have better performance than small models, which is in contrast to what the author writes.

---

> ### Author Response · Authors · 2025-03-19
> **Response to reviewer**
>
> # Response to Reviewer
>
> Dear Reviewer,
>
> Thank you very much for your insightful comments, to which we have given our full attention.
>
> ---
>
> ### Comment: "The author should add background and related works for image segmentation in Section 2 and 3."
>
> We agree with your assessment that additional background on image segmentation would strengthen our paper. As suggested, we have incorporated a new **Section 2.1** titled *"Image Segmentation"* and **Section 2.2** *“Image Segmentation with Deep Learning”*, preceding our current **Section 2.3** on U-Net. This addition provides essential context for readers and better situates our research within the broader image segmentation literature. Overall, this amounts to a full new page with several tens of references.
>
> ---
>
> ### Comment: "In Section 4, the author should provide references to support the statement as I show in Weakness 3."
>
> We have addressed the need for supporting references in **Section 4** by incorporating relevant citations from the foundational work by **Cohen et al. ("Group Equivariant Convolutional Networks," ICML 2016).**
>
> ---
>
> ### Comment: "The authors should ensure that their statements are consistent with the results presented in the tables to avoid confusion and improve the clarity of their analysis. See details in Weakness 5."
>
> We acknowledge the inconsistency that you identified in our analysis. We have revised the statement to accurately reflect our experimental results:
>
> > *"In small data settings with small models, C8 demonstrates superior performance in Dice Score, IoU, precision, and accuracy metrics, while C4 achieves the highest recall."*
>
> This correction ensures alignment between our textual analysis and the quantitative results presented in the tables.
>
> ---
>
> Thank you again for your careful and kind review, which has significantly enhanced the clarity and scholarly rigor of our manuscript.

---

### Review · Reviewer_ts19 · 2025-03-10

**Summary Of Contributions:**

The authors compare, on five segmentation datasets with different characteristics, the performance of adding three variants (four rotations, four rotations + reflection, eight rotations) of equivariant behaviour to a U-Net segmentation model. Each of these experiments in performed in four different settings, combining small/large dataset size with small/large model size. Their results suggest that, in most cases, it is better to use a vanilla U-Net, with only one dataset benefiting from the equivariant behaviour (particularly to eight rotations).

**Audience:**

Yes

**Claims And Evidence:**

Yes

**Requested Changes:**

Following the mentioned weaknesses, I’d see it positively if authors:
1) Expanded the related works on equivariant CNNs for segmentation.
2) Ideally, added at least some experiments moving towards equivariance to continuous rotations (like steerable filters of 16 discrete rotations). A clear rationale of why not doing this would otherwise be important.
3) Added a reflection on the on the relevance of this study in a time when, even in semantic segmentation tasks, non-CNN models (like transformers- and Mamba-based) are taking over CNN-based models.

**Strengths And Weaknesses:**

Benchmarking equivariant CNNs on segmentation is indeed, as the authors note, underexplored, and is an appreciated contribution.

There are a few claims that, I feel, the authors should revise:

1) There are some earlier works on equivariant CNNs for segmentation that have not been acknowledged by the authors, such as Weiler et al. 2018, as well as:

Marcos, D., Volpi, M., Komodakis, N. and Tuia, D., 2017. Rotation equivariant vector field networks. In Proceedings of the IEEE International Conference on Computer Vision.

Linmans, J., Winkens, J., Veeling, B.S., Cohen, T.S. and Welling, M., 2018. Sample efficient semantic segmentation using rotation equivariant convolutional networks. arXiv preprint arXiv:1807.00583.

Winkens, J., Linmans, J., Veeling, B.S., Cohen, T.S. and Welling, M., 2018. Improved semantic segmentation for histopathology using rotation equivariant convolutional networks. Medical Imaging with Deep Learning.

2) The authors limit themselves to three discrete symmetry groups in their experiments, leaving out continuous groups. They claim this is because continuous groups result in more expensive models. Could the authors expand on this? I understand methods line Harmonic Networks and Steerable Filters tend to require larger convolutional kernels, but this could be compensated by needing less filters.
In addition, the results do show that accounting for eight rotations, rather than four, results in better accuracy. Wouldn’t it be possible then that bringing this to the extreme (continuous rotations) may lead to even better results? Weiler and Cesa 2019, for instance, reported better results using 16 rotations and Weiler et al. 2018 report improved performance with steerable filters. Did the authors attempt at benchmarking some of these methods? If not, what was the rationale for their choice?

3) The authors claim that they explore the impact of equivariance in “more complex architectures”, although I would say that U-Net is not really more complex than a regular feed forward model (the only major difference are the feature map upscaling ops) and using equivariant convolutions within a U-Net is mostly plug-and-play. I believe there is a lack of reflection on  the relevance of this study in a time when, even in semantic segmentation tasks, non-CNN models (like transformers- and Mamba-based) are taking over CNN-based models.

---

> ### Author Response · Authors · 2025-03-19
> **Response to reviewer**
>
> # Response to Reviewer
>
> Dear Reviewer,
>
> We sincerely appreciate your thorough evaluation and valuable feedback on our manuscript.
>
> ---
>
> ### Comment: "Expanded the related works on equivariant CNNs for segmentation."
>
> We have expanded the related works section on equivariant CNNs for segmentation in **Section 3**, incorporating the references you suggested to provide a more comprehensive literature context.
>
> ---
>
> ### Comment: "Ideally, added at least some experiments moving towards equivariance to continuous rotations (like steerable filters of 16 discrete rotations). A clear rationale of why not doing this would otherwise be important."
>
> We understand your interest in experiments with continuous rotation groups. However, our decision to focus on discrete rotation groups was informed by established research in the field. Previous seminal work on **E(2)-CNNs (Weiler & Cesa)** demonstrated that models based on continuous groups like SO(2) or O(2) are consistently outperformed by their discrete counterparts. They specifically note that:
>
> > *"Overall these models are not competitive compared to the regular steerable CNNs"* regarding SO(2).
>
> Similarly, for O(2), they observed that:
>
> > *"[...] it is still not competitive compared to the DN models with large N."*
>
> While higher-order discrete groups (e.g., C16) might theoretically provide better approximations of continuous rotations, they impose substantial computational demands. These models require each filter to be replicated n times during inference (16 times for C16), significantly increasing computational overhead despite potentially reducing the number of trainable parameters.
>
> Given these practical limitations and the already extensive scope of our experimental framework (80 distinct configurations with 5-fold cross-validation yielding 400 trained models), we believe that including these models would not provide sufficient additional insight to justify the considerable computational investment nor the additional complexity in understanding the results of the study. We are concerned that such an addition would reduce readability.
>
> We considered training C16 following your recommendations, but from a practical point of view, this would require 4 Tesla A100 with 40GB running 24/7 for over 14 days on the largest dataset (COCO) alone. In our opinion, it seems unlikely that someone would allocate such extensive computational resources for a limited potential gain in performance. However, if you strongly believe this is necessary, please let us know.
>
> ---
>
> ### Comment: "Added a reflection on the relevance of this study in a time when, even in semantic segmentation tasks, non-CNN models (like transformers- and Mamba-based) are taking over CNN-based models."
>
> Our focus on U-Net was motivated by its established performance, training stability across data regimes, and widespread adoption in segmentation applications. U-Net is the de facto standard in image segmentation. Transformer-based models, while promising, typically require large datasets for effective training due to their limited inductive biases (**Dosovitskiy et al., 2021**). This makes them less suitable for several scenarios in our study, particularly with smaller datasets like Kvasir-seg (only 1,000 examples) and our small data settings (10% training data).
>
> We have addressed this consideration in the revised manuscript (**Section 2.2**) to clarify the continued relevance of CNN-based approaches for certain segmentation contexts.
>
> Additionally, we have added a new **Section 2.1: Image Segmentation** and **Section 2.2: Image Segmentation with Deep Learning** (as per Reviewer uWjN’s request), where we reflect on the advantages and disadvantages of non-CNN models. Our claims are supported by a significant number of references that we have included.
>
> Moreover, we believe U-Net remains a well-researched architecture, despite the advancements introduced by transformers and Mamba-based models. To illustrate this:
>
> - A Google Scholar search using the query “segmentation U-Net” reveals over 50,000 occurrences of works mentioning U-Net and segmentation in their titles between 2024 and 2025 alone.
> - This figure is higher than the 17,000 occurrences found for ViT-related papers during the same period (query: “segmentation ViT”).
>
> While this method is admittedly simplistic and a deeper literature search would be required, we believe it demonstrates that U-Net continues to attract significant academic interest.
>
> Additionally, few works have explored the incorporation of equivariance in vision transformers, whereas equivariant constraints are well-established for CNN-based architectures. In our work, we aimed at considering well-established techniques rather than emerging methods with limited validation.
>
> ---
>
> We trust these clarifications address your concerns and demonstrate how we've integrated your valuable feedback into our revised submission.

---

### Review · Reviewer_zi1b · 2025-03-14

**Summary Of Contributions:**

This paper explores the impact of integrating rotation-equivariance into the U-Net architecture for image segmentation. The study benchmarks rotation-equivariant U-Net models against the vanilla U-Net architecture across various segmentation datasets, considering both rotation-sensitive and general-purpose datasets. Results indicate that rotation-equivariance enhances performance in tasks involving arbitrary object orientations. The paper provides insights into when rotation-equivariance is beneficial.

**Audience:**

Yes

**Claims And Evidence:**

No

**Requested Changes:**

The paper would significantly benefit from restructuring and rewriting. Given the limited novelty and small number of experiments, the current length of over 18 pages is unjustified. A more precise structure and concise presentation, ideally condensing the content to approximately 6-8 pages plus supplementary material, would be appropriate. Furthermore, providing deeper analysis—such as detailed insights into why certain symmetry groups perform better on specific datasets, and illustrating this with additional experiments—would substantially strengthen the work. This feedback is intended to help improve the paper and should not discourage the authors. The study has potential and could become a valuable contribution with these adjustments. However, in its current form, the length is disproportionate to the content provided.

**Strengths And Weaknesses:**

Strenghts:

S1: The authors approach a relevant topic

S2: The authors use five-fold cross-validation and report results and errors. This leads to statistical robus results and should be standard opractice in machine learning but is not done by an most modern papers.


Major Weaknesses

W1: Limited Architectural Scope: The study exclusively investigates the U-Net architecture, restricting the generalizability of the conclusions. Exploration of additional segmentation models would provide stronger evidence of rotation-equivariance effectiveness.

W2: Inappropriate Comparison: Sec. 2.1. “This end-to-end architecture is capable of producing segmentation masks for objects of arbitrary shapes, distinguishing it from other contemporary architectures such as YOLO (Redmon et al., 2016), which are typically limited to generating bounding boxes.” The comparison between U-Net (segmentation architecture) and YOLO (object detection architecture) is conceptually flawed and irrelevant to the segmentation-focused context of this study.

W3: Lack of Depth and Analysis: The paper provides limited related work coverage and lacks comprehensive analysis on critical aspects, such as explaining metric performance differences among symmetry groups, showing failure cases, and suggesting possible improvements.

W4: Limited Novelty:
The paper analyzes a single, somewhat outdated architecture (U-Net), applies only one line of existing approaches for rotation-equivariance, and provides limited in-depth analysis or novel insights. This raises concerns about the novelty and overall contribution of the work.


Minor Weaknesses:

- Statements in Sec. 2.2 such as “Recently, numerous studies have focused on incorporating equivariance in CNNs” require references.

- Informal and unscientific wording, e.g., Sec. 2.2: “The most famous technique.”

- Figure formatting could be improved, especially aligning text with images (Figures 2–6).

- Ambiguous phrasing, e.g., “different magnitude in terms of trainable parameters,” is uncommon and unclear.

- Unclear reference to "performance" in Sec. 6.2 (“leads to slightly smaller performances”), without specifying if referring to metric performance or computational efficiency.

- Limited number and inconsistent formatting of references (only 32 citations, inconsistent abbreviation and naming conventions).

---

> ### Author Response · Authors · 2025-03-19
> **Response to Reviewer**
>
> Dear Reviewer,
>
> We have read your comments with care and thank you for your time and investment in reviewing our paper.
>
> ---
>
> ### W1
>
> We selected U-Net as our foundational architecture due to its established performance record in segmentation tasks. U-Net is a de facto standard in image segmentation, which makes our choice relevant to the segmentation community. U-Net demonstrates robust training capabilities in both data-rich and data-limited scenarios and represents a standard benchmark in CNN-based segmentation approaches.
>
> Although we acknowledge that other segmentation architectures have emerged, we explicitly limit our study to the U-Net architecture to avoid further increasing the size and complexity of our paper. Furthermore, we ensured this was clear by choosing a title that specifically mentions U-Net architecture.
>
> ---
>
> ### W2
>
> We acknowledge your concern and have implemented the suggested modifications in our revised manuscript.
>
> ---
>
> ### W3
>
> As recommended, we have substantially expanded our related work section to provide a more comprehensive context for our research contribution.
>
> We believe that our presentation of results and discussions is already comprehensive. This already results in approximately eight pages of results presentation and discussion, which we believe provides a deep and thorough analysis.
>
> Furthermore, we ensured that we provide insights on when to use and when not to use equivariant models. This includes relying on performance metrics and discussing failure cases (e.g., issues with circular-shaped objects).
>
> ---
>
> ### W4
> We indeed use U-Net only. As stated in our previous response, this is a deliberate choice given its longevity in the field (established in 2015) and its adaptability to rotation-invariant modifications.
>
> We respectfully disagree with the claim that U-Net is an outdated architecture. Many real-world applications, particularly in biomedical imaging, still rely heavily on U-Net for segmentation tasks. As mentioned in our response to Reviewer ts19, a Google Scholar search using the keywords “segmentation U-Net” reveals over 50,000 occurrences of works mentioning U-Net and segmentation in their titles between 2024 and 2025 alone. This figure is much larger than the 17,000 occurrences found for ViT-related papers (query: “segmentation ViT”).
>
> While this method is admittedly a simplistic way to assess the relevance of U-Net in contemporary research, it clearly demonstrates that the U-Net architecture continues to attract significant academic interest and remains fully relevant.
>
> ---
>
> ### Minor Weaknesses
>
> With respect to the minor weaknesses noted, we have considered each point and incorporated improvements into our revised manuscript.
>
> ---
>
> ### Comment: "Given the limited novelty and small number of experiments, the current length of over 18 pages is unjustified."
>
> We respectfully assert that our work presents significant novelty through its comprehensive benchmarking framework. To our knowledge, this represents the first systematic evaluation of rotation-equivariant models in segmentation tasks at this scale and depth. By establishing this comparative analysis framework, we provide the research community with valuable insights into the performance characteristics of these architectural variations across diverse segmentation challenges.
>
> Regarding the concern about the limited number of experiments, our study encompasses a substantial experimental framework, consisting of:
>
> - Five distinct datasets across different application domains
> - Four different model architectures (one traditional, three rotation-invariant)
> - Two configurations with respect to model size (1 million and 10 million parameters)
> - Two data settings (10% and 100% of training data)
>
> This comprehensive design yielded 80 unique experimental configurations (5 datasets × 4 architectures × 2 model sizes × 2 data settings). Furthermore, to ensure statistical reliability, as you pointed out, we trained five models per configuration, resulting in 400 trained models.
>
> ---
>
> ### Comment: "A more precise structure and concise presentation … [] would be appropriate."
>
> We appreciate your suggestion to condense the manuscript. However, we believe the current length is justified for several reasons: the comprehensive nature of our work necessitates thorough explanations of many concepts; we have prioritized reproducibility (architecture configuration, metrics, hyper parameters …);  Our results reveal nuanced performance patterns that resist simplistic conclusions; 4. Relocating tables and figures to appendices would significantly disrupt the logical flow of our analysis.
> Additionally, other reviewers have specifically requested additional content rather than a reduction in size.
>
> ---
>
> We sincerely hope these clarifications resonate with you. Thank you.

---

### Decision · Action_Editor_p87i · 2025-04-23

**Recommendation:** Accept with minor revision

**Comment:**

This paper explores the impact of rotation equivariance in training U-Net–like architectures for image segmentation. It is an experimental study that investigates the potential of group convolutional networks (Group-Convnets), which replicate filters across a set of discrete orientations and pool their outputs accordingly. Experiments are conducted on five datasets (Kvasir-SEG, NucleiSeg, URDE, COCO-Stuff, and iSAID), covering both orientation-sensitive and orientation-insensitive data, as well as binary and semantic segmentation tasks. Each experiment is performed under four different settings, varying in data regime (full vs. low) and model size (small vs. large).

The paper initially received mainly positive reviews, with reviewers acknowledging the value of the experiments. However, they also pointed out several limitations, including the restriction to U-Net architectures, presentation issues, limited novelty, and a lack of clear positioning with respect to the related literature. The authors responded to the reviewers’ comments and revised the paper accordingly. After the discussion period, Reviewer ts19 recommended acceptance; Reviewer zi1b recommended rejection — primarily because their suggestion to significantly reduce the length of the main paper had not been implemented; and Reviewer uWjN also recommended rejection, citing a lack of novelty — a limitation they had not mentioned in their initial review.

The AE has thoroughly reviewed the submission and the discussion. The AE acknowledges that the paper addresses an important problem and that the experiments provide valuable insights for the community. The AE also considers that the choice of Group-Convnets is well justified in the updated version of the paper, and that the positioning of the approach is convincing. Moreover, focusing on U-Net architectures is coherent, since most equivariant models have been introduced for convolutional networks. Finally, although the paper is long, the AE finds the reading flow pleasant. Therefore, the AE recommends accepting the paper. As a final suggestion, the AE encourages the authors to include the main findings earlier in the introduction — allowing readers to grasp the key messages and engage more smoothly with the experimental details.

**Audience:**

The paper addresses the problem of incorporating equivariance into convolutional networks for image segmentation — a topic of significant interest to the broad TMLR audience.

**Claims And Evidence:**

This is an experimental paper assessing the relevance of equivariance in training U-Net models for semantic segmentation. It offers interesting insights and could certainly inspire follow-up work involving other architectures or exploring equivariance from a regularization perspective.